# POSE MODULATED AVATARS FROM VIDEO

**Chunjin Song[1], Bastian Wandt[2] & Helge Rhodin[1,3]**
[1]Department of Computer Science, University of British Columbia
[2]Department of Electrical Engineer, Linköping University
[3]Bielefeld University
{chunjins,rhodin}@cs.ubc.ca
bastian.wandt@liu.se

## ABSTRACT

It is now possible to reconstruct dynamic human motion and shape from a sparse set of cameras using Neural Radiance Fields (NeRF) driven by an underlying skeleton. However, a challenge remains to model the deformation of cloth and skin in relation to skeleton pose. Unlike existing avatar models that are learned implicitly or rely on a proxy surface, our approach is motivated by the observation that different poses necessitate unique frequency assignments. Neglecting this distinction yields noisy artifacts in smooth areas or blurs fine-grained texture and shape details in sharp regions. We develop a two-branch neural network that is adaptive and explicit in the frequency domain. The first branch is a graph neural network that models correlations among body parts locally, taking skeleton pose as input. The second branch combines these correlation features to a set of global frequencies and then modulates the feature encoding. Our experiments demonstrate that our network outperforms state-of-the-art methods in terms of preserving details and generalization capabilities. Our code is available at https://github.com/ChunjinSong/PM-Avatars.

## 1 INTRODUCTION

Human avatar modeling has garnered significant attention as enabling 3D telepresence and digitization with applications ranging from computer graphics (Wu et al., 2019; Bagautdinov et al., 2021; Peng et al., 2021a; Lombardi et al., 2021) to medical diagnosis (Hu et al., 2022). To tackle this challenge, the majority of approaches start from a skeleton structure that rigs a surface mesh equipped with a neural texture (Bagautdinov et al., 2021; Liu et al., 2021) or learnable vertex features (Kwon et al., 2021; Peng et al., 2021a;b). Although this enables reconstructing intricate details with high precision (Liu et al., 2021; Thies et al., 2019) in controlled conditions, artifacts remain when learning the pose-dependent deformation from sparse examples. To counteract, existing methods typically rely on a parametric template obtained from a large number of laser scans, which still limits the variety of the human shape and pose. Moreover, their explicit notion of vertices and faces is difficult to optimize and can easily lead to foldovers and artifacts from other degenerate configurations. Moreover, the processed meshes are usually sampled uniformly in one static pose thus not being adaptive to dynamic shape details like wrinkles.

Our objective is to directly reconstruct human models with intricate and dynamic details from given video sequences with a learned neural radiance field (NeRF) model (Mildenhall et al., 2020). Most related are surface-free approaches such as A-NeRF (Su et al., 2021) and NARF (Noguchi et al., 2021) that directly transform the input query points into relative coordinates of skeletal joints and then predict density and color for volumetric rendering without an intermediate surface representation. To further enhance the ability to synthesize fine details, (Su et al., 2022; Li et al., 2023) explicitly decompose features into local part encodings before aggregating them to the final color. Closely related are also methods that learn a neural radiance field of the person in a canonical T-pose (Jiang et al., 2022; Li et al., 2022a; Wang et al., 2022). Despite their empirical success, as depicted in Fig. 1, it is evident that a single query point, when considered in different pose contexts, is difficult to be learned. Specifically, the T-shirt region appears flat in one pose ($2^{nd}$ row) but transforms into a highly textured surface in another ($1^{st}$ row). The commonly used positional

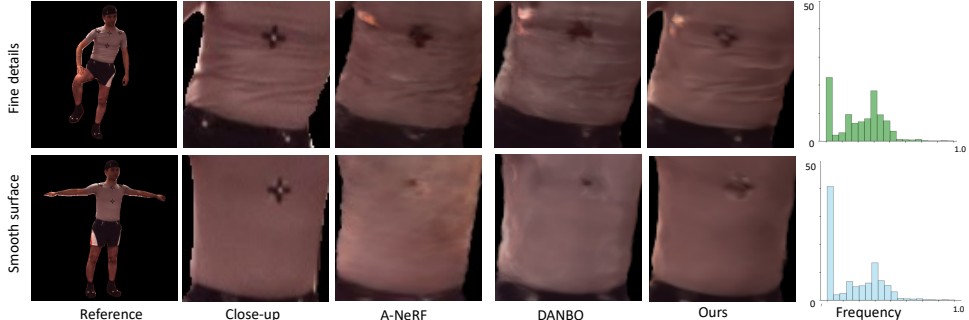

Figure 1: **Motivation.** Our frequency modulation approach enables pose-dependent detail by using explicit frequencies that depend on the pose context, varying over frames and across subjects. Our mapping mitigates artifacts in smooth regions as well as synthesizes fine geometric details faithfully, e.g., when wrinkles are present ($1^{st}$ row). By contrast, existing surface-free representations struggle either with fine details (e.g. marker) or introduce noise artifacts in uniform regions (e.g. the black clouds, $2^{nd}$ row). To quantify the difference in frequency of these cases, we calculate the standard deviation (STD) pixels within $5 \times 5$ patches of the input closeup and illustrate the frequency histograms of the reference. Even for the same subject in similar pose the frequency distributions are distinct, motivating our pose-dependent frequency formulation.

encoding (Mildenhall et al., 2020) maps points with fixed frequency transformations and is hence non-adaptive. Entirely implicit mappings from observation to canonical space are complex and contain ill-posed one to many settings. Thus they struggle to explicitly correlate pose context with the matching feature frequency bands of query points. As a result, these methods either yield overly smoothed details or introduce noisy artifacts in smooth regions.

In this paper, we investigate new ways of mapping the skeleton input to frequencies of a dynamic NeRF model to tackle the aforementioned challenges. Specifically, we design a network with a branch that is explicit in the frequency space and build a multi-level representation that adapts to pose dependencies. To accomplish this, we utilize Sine functions as activations, leveraging its explicit notion of frequency and its capability to directly enforce high-frequency feature transformations (Sitzmann et al., 2020; Mehta et al., 2021; Wu et al., 2023). The main challenge that we address here is on how to control the frequency of the Sine activation for deforming characters.

We first apply a graph neural network (GNN) (Su et al., 2022) on the input pose to extract correlations between skeleton joints, thereby encoding pose context. Given query point coordinates for NeRF rendering, the joint-specific correlation features are first combined with a part-level feature aggregation function and then utilized to generate point-dependent frequency transformation coefficients. The frequency modulation process takes place within a set of intermediate latent features, allowing us to optimize the modulation effects at different scales effectively. Lastly, similar to existing NeRF methods, we output density and color to synthesize and render images. Across various scenarios, we consistently outperform state-of-the-art methods. Our core contributions are

- We introduce a novel and efficient neural network with two branches, tailored to generate high-fidelity functional neural representations of human videos via frequency modulation.

- A simple part feature aggregation function that enables high-frequency detail synthesis in sharp regions and reduces artifacts near overlapping joints.

- We conduct thorough evaluation and ablation studies, which delve into the importance of window functions and frequency modulations with state-of-the-art results.

## 2 RELATED WORK

Our work is in line with those that apply neural fields to model human avatar representations. Here we survey relevant approaches on neural field (Xie et al., 2022) and discuss the most related neural avatar modeling approaches.

**Neural Fields.** As a successful application, a breakthrough was brought by Neural Radiance Fields (NeRF) (Mildenhall et al., 2020). Recently, extending NeRF to dynamic scenes becomes more and more popular and enables numerous downstream applications (Park et al., 2021a; Pumarola et al., 2021; Park et al., 2021b; Cao & Johnson, 2023). The key idea is to either extend NeRF with an additional time dimension (T-NeRF) and additional latent code (Gao et al., 2021; Li et al., 2022b; 2021b), or to employ individual multi-layer perceptrons (MLPs) to represent a time-varying deformation field and a static canonical field that represents shape details (Du et al., 2021; Park et al., 2021a;b; Tretschk et al., 2021; Yuan et al., 2021). However, these general extensions from static to dynamic scenes only apply to small deformations and do not generalize to novel input poses.

Our method is also related to applying periodic functions for high frequency detail modeling within the context of neural fields. NeRF (Mildenhall et al., 2020) encodes the 3D positions into a high-dimensional latent space using a sequence of fixed periodic functions. Later on, Tancik *et al.* (Tancik et al., 2020) carefully learn the frequency coefficients of these periodic functions, but they are still shared for the entire scene. In parallel, Sitzmann *et al.* (Sitzmann et al., 2020) directly uses a Sine-function as the activation function for latent features, which makes frequency bands adaptive to the input. (Lindell et al., 2022; Fathony et al., 2021) further incorporate the multi-scale strategy of spectral domains to further advance the modeling of band-limited signals. Recently, (Hertz et al., 2021; Mehta et al., 2021; Wu et al., 2023) propose to modulate frequency features based on spatial patterns for better detail reconstructions. However, differing from these methods, we explicitly associate the desired frequency transformation coefficients with pose context, tailored to the dynamics in human avatar modeling.

**Neural Fields for Avatar Modeling.** Using neural networks to model human avatars (Loper et al., 2015) is a widely explored problem (Deng et al., 2020; Saito et al., 2021; Chen et al., 2021). However, learning personalized body models given only videos of a single avatar is particularly challenging which is our research scope in this paper.

In the pursuit of textured avatar modeling, the parametric SMPL body model is a common basis. For instance, (Zheng et al., 2022; 2023) propose partitioning avatar representations into local radiance fields attached to sampled SMPL nodes and learning the mapping from SMPL pose vectors to varying details of human appearance. On the other hand, approaches without a surface prior, such as A-NeRF (Su et al., 2021) and NARF (Noguchi et al., 2021), directly transform the input query points into relative coordinates of skeletal joints. Later on, TAVA (Li et al., 2022a) jointly models non-rigid warping fields and shading effects conditioned on pose vectors. ARAH (Wang et al., 2022) explores ray intersection on a NeRF body model initialized using a pre-trained hypernetwork. Similar to former disentanglement (Wu et al., 2019; Song et al., 2019; Wu et al., 2020; 2022), DANBO (Su et al., 2022) applies a graph neural network to extract part features and decompose an independent part feature space for a scalable and customizable model. To reconstruct high-frequency details, Neural Actor (Liu et al., 2021) utilizes an image-to-image translation network to learn texture mapping, with a constraint on performers wearing tight clothes for topological consistency. Further studies (Peng et al., 2021b;a; Dong et al., 2022) suggest assigning a global latent code for each training frame to compensate for dynamic appearance. Most recently, HumanNeRF (Weng et al., 2022) and its following works like Vid2Avatar (Guo et al., 2023) and MonoHuman (Yu et al., 2023) show high-fidelity avatar representations for realistic inverse rendering from a monocular video. Despite the significant progress, none of them explicitly associate the pose context with frequency modeling, which we show is crucial for increasing shape and texture detail.

## 3 METHOD

Our objective is to reconstruct a 3D animatable avatar by leveraging a collection of $N$ images, along with the corresponding body pose represented as the sequence of joint angles $[\theta_k]_{k=1}^N$. Our key technical ingredient is a pose-guided frequency modulation network and its integration for avatar reconstruction. Fig. 2 provides a method overview with three main components. First, we employ a Graph Neural Network (GNN) to estimate local relationships between different body parts. The GNN facilitates effective feature aggregation across body parts, enabling to learn the nearby pose contexts without relying on surface priors. Then the aggregated GNN features are learned to modulate the frequencies of input positions. Lastly, the resulting per-query feature vector is mapped to the corresponding density and radiance at that location as in the original NeRF framework.

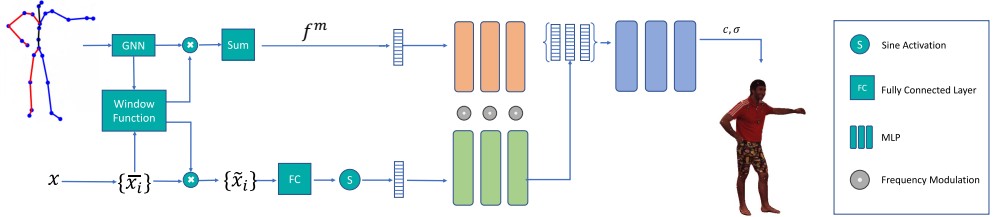

Figure 2: **Method overview.** First, a graph neural network takes a skeleton pose as input to encode correlations of joints. Together with the relative coordinates $\{\bar{x}_i\}$ of query point $x$, a window function is learned to aggregate the features from all parts. Then the aggregated GNN features are used to compute frequency coefficients (orange) which later modulate the feature transformation of point $x$ (green). Finally, density $\sigma$ and appearance $c$ is predicted as in NeRF.

## 3.1 PART-RELATIVE POSE ENCODING

Inspired by DANBO (Su et al., 2022), we adopt a graph representation for the human skeleton, where each node corresponds to a joint that is linked to neighboring joints by bones. For a given pose with $N_B$ joints $\theta = [\omega_1, \omega_2, \ldots, \omega_{N_B}]$, where $\omega_i \in \mathbb{R}^6$ (Zhou et al., 2019) is the rotation parameter of bone $i \in \{1, 2, \ldots, N_B\}$, we regress a feature vector for each bone part as:

$$[G_1, G_2, \ldots, G_{N_B}] = \text{GNN}(\theta), \tag{1}$$

where GNN represents a learnable graph neural network. To process the irregular human skeleton, we employ two graph convolutional layers, followed by per-node 2-layer Multi-Layer Perceptrons (MLPs). To account for the irregular nature of human skeleton nodes, we learn individual MLP weights for each node; see (Su et al., 2022) for more comprehensive details.

Given a sample location $x \in \mathbb{R}^3$ in global coordinates for which the NeRF should output color and density, we first map it to the $i$-th bone-relative space as

$$\begin{bmatrix} \hat{x}_i \\ 1 \end{bmatrix} = T(\omega_i) \begin{bmatrix} x \\ 1 \end{bmatrix}, \tag{2}$$

where $T(\omega_i)$ denotes the world-to-bone coordinates transformation matrix computed by the rotation parameter $\omega_i$. We first perform a validness test for the scaled relative positions $\{\bar{x}_i = s_i \cdot \hat{x}_i\}$, where $s_i$ is a learnable scaling factor to control the size of the volume the $i$-th part contributes to. This facilitates the processing efficiency and concentrates the network on local patterns. If no $\{\bar{x}_i\}$ falls in $[-1, 1]$, $x$ is estimated to locate far from the body surface and is discarded. Then these features are employed to drive the frequency modulation adaptively.

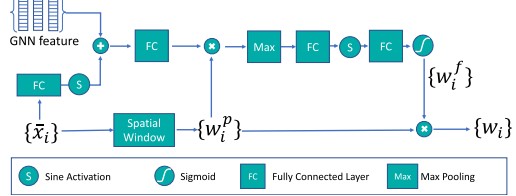

Figure 3: **Learned window function**. The query point location is processed with a spatial and pose-dependent window to remove spurious correlations between distant joints.

## 3.2 FREQUENCY MODULATION

**Two-stage Window Function.** To decide which per-part point-wise features to pass on to the downstream network and address the feature aggregation issues in (Su et al., 2022), we design a learnable two-stage window function. In Fig. 3, the window function takes the per-part GNN feature set $\{G_i\}$ and the scaled relative position set $\{\bar{x}_i\}$ of a valid point $x$ as inputs. To facilitate learning volume dimensions $\{s_i\}$ that adapt to the body shape and to mitigate seam artifacts, we define

$$w_i^p = \exp(-\alpha(\|\bar{x}_i\|_2^\beta)), \tag{3}$$

where $\|\cdot\|_2$ is the $L_2$-norm. The $w_i^p$ function attenuates the extracted feature based on the relative spatial distances to the bone centers such that multiple volumes are separated via the spatial similarities between $x$ and the given parts. Thus we name $w_i^p$ as the spatial window function. We set $\alpha = 2$ and $\beta = 6$ as in (Lombardi et al., 2021). However, it is possible that one part might prioritize

over other parts when multiple valid parts' feature space overlap. This motivates us to take the pose context into account and consider the point-wise feature of each part as the second stage. First, we perform $\dot{x}_i = \sin(\mathbf{W}_c \cdot \bar{x}_i)$ for each $\bar{x}_i$, where $\mathbf{W}_c$ indicates a Gaussian initialized fully-connected layer as in FFN (Tancik et al., 2020). After concatenating $\{\dot{x}_i\}$ and the GNN features $\{G_i\}$ correspondingly, we apply a full-connected layer to regress $\{f_i^p\}$ as the point-wise feature of the $i$-th part. Then we attenuate the feature as $f_i^w = f_i^p \cdot w_i^p$ to focus on spatially nearby parts.

To further decide which part $x$ belongs to, we compute relative weights by aggregating all $\{f_i^w\}$ through a max pooling for the holistic shape-level representation. The max-pooled feature is fed into a sequence of fully-connected layers which are activated by a Sigmoid function to output a per-part weight $w_i^f$. We call the feature window function $w_i^f$ to echo the spatial window function $w_i^p$ since it operates on pose features. Finally, we compute the per-part weight $w_i = w_i^p \cdot w_i^f$ and output modulation frequencies as

$$f^m = \sum_{i=0}^{N_B} w_i \cdot G_i, \quad [\theta_1, \theta_2, \cdots, \theta_n] = \text{MLP}\,(f^m), \tag{4}$$

where $f^m$, $\theta_i$ and $n$ represent the aggregated part feature, the modulation frequency coefficient at $i$-th layer and the layer number of the subsequent modulation module, respectively.

After the two-stage window function, the extracted $f^m$ sparsely correspond to a small part set. To echo the locality assumption throughout the entire network, we also perform the window function $w_i$ for the $N_B$ relative positions of query point $x$ as $\{\tilde{x}_i = \bar{x}_i \cdot w_i\}$. Note that, we aggregate the part features before MLP-based frequency modulation to avoid time-consuming processing over all parts for all samples. Thus computation complexity reduces significantly.

**Frequency Prediction.** Inspired by pi-GAN (Chan et al., 2021), we build the backbone network for each $x$ with a series of Sine-activated fully-connected layers (Sitzmann et al., 2020). To this end, we first concatenate all the re-weighed positions $\{\tilde{x}_i\}$ as a whole and input it into a Sine-activated fully-connected layer (Sitzmann et al., 2020). Later on, each fully-connected layer is defined as

$$\mathbf{f}_i = \sin(\theta_i \cdot \mathbf{W}_i \mathbf{f}_{i-1} + \mathbf{b}_i), \tag{5}$$

where $\mathbf{W}_i$ and $\mathbf{b}_i$ are trainable weight and bias in the $i$-th layer $L_i$. Finally, we concatenate the Sine-activated features $\{\mathbf{f}_i\}$ as $S(x) = [\mathbf{f}_1, \mathbf{f}_2, \cdots, \mathbf{f}_n]$ for further processing.

**Design Discussions and relation to DANBO.** Both our method and DANBO (Su et al., 2022) use the GNN features as a building block to measure bone correlations. By contrast to DANBO which directly estimates part-level feature spaces from GNN features, we leverage these aggregated GNN features to estimate the appropriate frequencies, driving the frequency modulation for the input positions. This enables the linked MLP networks to adaptively capture a wide spectrum of coarse and fine details with high variability, as illustrated in the visual comparisons to DANBO in Fig. 1, 4, 5, 6. Moreover, although some work (Hertz et al., 2021; Wu et al., 2023) aims to modulate frequency features with locality, we make the first step for pose-dependent frequency modulation, which is critical in human avatar modeling. We start from existing conceptual components, analyze their limitations, and propose a novel solution. Specifically, we focus on how to connect GNN pose embeddings with frequency transformations. We then propose a simple window function to improve efficiency without losing accuracy which is also special and helpful in neural avatar modeling.

### 3.3 VOLUME RENDERING AND LOSS FUNCTIONS

The output feature $S(x)$ can accurately capture the information of pose dependency and spatial positions, and thus enables adaptive pattern synthesis. To obtain high-quality human body, we learn a neural field $F$ to predict the color $c$ and density $\sigma$ at position $x$ as

$$(\mathbf{c}, \sigma) = F(S(x), r), \tag{6}$$

where $r \in \mathbb{R}^2$ indicates the given ray directions. Following the existing neural radiance rendering pipelines for human avatars (Su et al., 2021; 2022; Wang et al., 2022), we output the image of the human subject as in the original NeRF:

$$\hat{C}\,(r) = \sum_{i=1}^{n} \mathcal{T}_i \,(1 - \exp(-\sigma_i \delta_i))\,\mathbf{c}_i, \mathcal{T}_i = \exp(-\sum_{j=1}^{i-1} \sigma_j \delta_j). \tag{7}$$

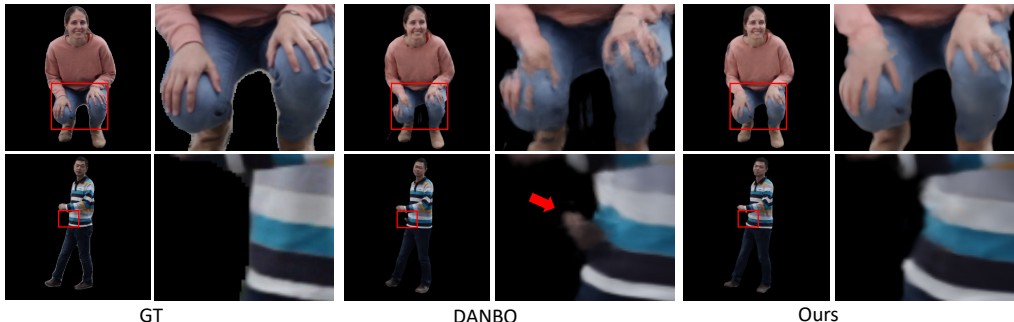

Figure 4: **Visual comparisons on MonoPerfCap.** We can preserve better shape contours ($1^{st}$ row) and produce realistic cloth textures without artifacts (highlighted by the red arrow on $2^{nd}$ row).

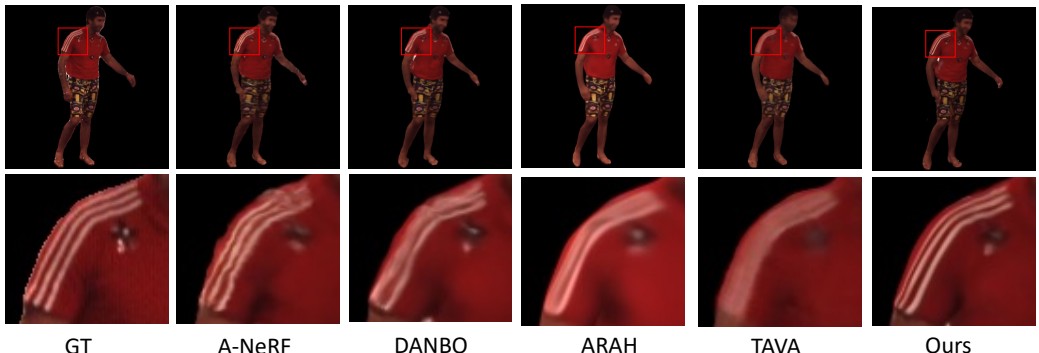

Figure 5: **Visual comparisons for novel view synthesis.** Compared to baselines, we can vividly reproduce the structured patterns.

Here, $\hat{C}$ and $\delta_i$ indicate the synthesized image and the distance between adjacent samples along a given ray respectively. Finally, we compute the $L_1$ loss $\|\cdot\|_1$ for training as

$$\mathcal{L}_{rec} = \sum_{\mathbf{r} \in \mathfrak{R}} \left\| \hat{C}(r) - C_{gt}(r) \right\|_1, \tag{8}$$

where $\mathfrak{R}$ is the whole ray set and $C_{gt}$ is the ground truth. The usage of $L_1$ loss is to enable more robust network training. Following (Lombardi et al., 2021), we add a regularization loss on the scaling factors to prevent the per-bone volumes from growing too large and taking over other volumes:

$$\mathcal{L}_{\mathrm{s}} = \sum_{i=1}^{N_B} (s_i^x \cdot s_i^y \cdot s_i^z), \tag{9}$$

where $\{s_i^x, s_i^y, s_i^z\}$ are the scaling factors along $\{x, y, z\}$ axes respectively. Hence, our total loss with weight $\lambda_s$ is

$$\mathcal{L} = \mathcal{L}_{rec} + \lambda_s \mathcal{L}_{\mathrm{s}}. \tag{10}$$

## 4 RESULTS

In this section, we compare our approach with several state-of-the-art methods, including Neural-Body (Peng et al., 2021b), Anim-NeRF (Peng et al., 2021a), A-NeRF (Su et al., 2021), TAVA (Li et al., 2022a), DANBO (Su et al., 2022), and ARAH (Wang et al., 2022). These methods vary in their utilization of surface-free, template-based, or scan-based priors. We also conduct an ablation study to assess the improvement achieved by each network component. This study analyzes and discusses the effects of the learnable window function. Source code will be released with the publication.

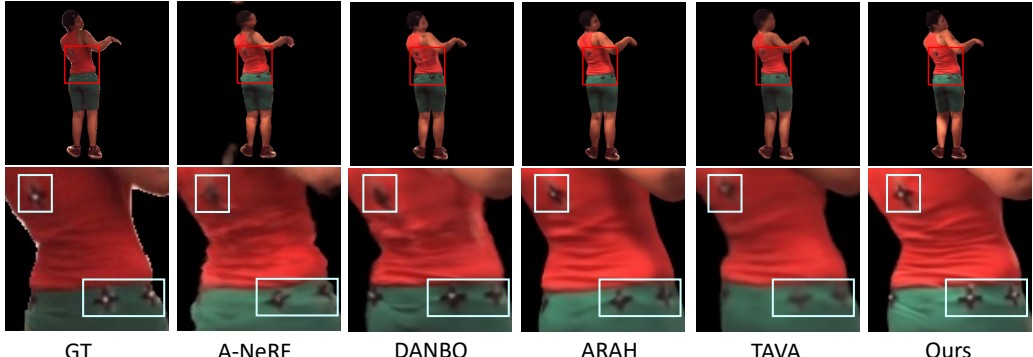

GT  A-NeRF  DANBO  ARAH  TAVA  Ours

Figure 6: **Visual comparisons for novel pose rendering.** For novel poses that are unseen during training, cloth wrinkles form chaotically. Hence, none of the methods is expected to match the folds. Ours yields the highest detail, including the highlighted marker texture.

Table 1: **(a) Unseen pose synthesis on MonoPerfCap Xu et al. (2018).** Our full model shows better overall perceptual quality over chosen models from monocular videos. **(b) Ablation studies on Human3.6M S9.** Our full model outperforms all ablated baselines across all metrics.

|         | ND | | WP | | Avg | |
|---------|--------|--------|--------|--------|--------|--------|
|         | PSNR↑ | SSIM↑ | PSNR↑ | SSIM↑ | PSNR↑ | SSIM↑ |
| onlyGNN | 20.03 | 0.841 | 22.71 | 0.863 | 21.37 | 0.852 |
| onlySyn | 19.57 | 0.827 | 22.66 | 0.860 | 21.12 | 0.844 |
| DANBO   | 20.10 | 0.842 | 22.39 | 0.861 | 21.25 | 0.852 |
| **Ours** | **20.55** | **0.853** | **22.85** | **0.866** | **21.70** | **0.860** |

(a)

|            | onlyGNN | noGNN | onlySyn | $only\ w_i^p$ | $only\ w_i^f$ | $no\ window$ | Ours (full) |
|------------|---------|-------|---------|---------------|---------------|--------------|-------------|
| **Novel view** | | | | | | | |
| PSNR↑      | 25.91 | 26.08 | 25.86 | 26.21 | 25.91 | 22.21 | **26.39** |
| SSIM↑      | 0.917 | 0.925 | 0.916 | 0.925 | 0.921 | 0.639 | **0.929** |
| LPIPS↓     | 0.120 | 0.105 | 0.118 | 0.112 | 0.124 | 0.478 | **0.100** |
| **Novel pose** | | | | | | | |
| PSNR↑      | 24.75 | 24.45 | 24.82 | 25.01 | 24.72 | 20.98 | **25.12** |
| SSIM↑      | 0.891 | 0.899 | 0.901 | 0.904 | 0.900 | 0.628 | **0.908** |
| LPIPS↓     | 0.146 | 0.131 | 0.141 | 0.133 | 0.148 | 0.502 | **0.122** |

(b)

## 4.1 EXPERIMENTAL SETTINGS

We evaluate our method on widely recognized benchmarks for body modeling. Following the protocol established by Anim-NeRF, we perform comparisons on the seven actors of the Human3.6M dataset (Ionescu et al., 2011; 2013; Peng et al., 2021a) using the method described in (Gong et al., 2018) to compute the foreground maps. Like DANBO, we also apply MonoPerfCap (Xu et al., 2018) as a high-resolution dataset to evaluate the robustness to unseen poses in monocular videos.

To ensure a fair comparison, we follow the previous experimental settings including the dataset split and used metrics (Su et al., 2021; 2022; Li et al., 2022a). Specifically, we utilize standard image metrics such as pixel-wise Peak Signal-to-Noise Ratio (PSNR) and Structural Similarity Index Metric (SSIM) (Wang et al., 2004) to evaluate the quality of output images. Additionally, we employ perceptual metrics like the Learned Perceptual Image Patch Similarity (LPIPS) (Zhang et al., 2018) to assess the structural accuracy and textured details of the generated images. Since our primary focus is on the foreground subjects rather than the background image, we report the scores based on tight bounding boxes, ensuring the evaluation to be focused on the relevant regions of interest.

## 4.2 NOVEL VIEW SYNTHESIS

To evaluate the generalization capability under different camera views, we utilize multi-view datasets where the body model is learned from a subset of cameras. The remaining cameras are then utilized as the test set, allowing us to render the same pose from unseen view directions.

We present the visual results in Fig. 5. Comparing to the selected baselines, our method shows superior performance in recovering fine-grained details, as evident in the examples such as the stripe texture depicted on the first row. We attribute this to the explicit frequency modulation which mitigates grainy artifacts and overly smooth regions. Tab. 2 quantifies our method's empirical advantages, supporting our previous findings.

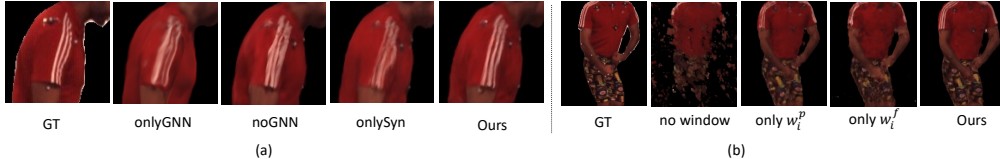

Figure 7: **Ablation studies** on sub-branch networks (a) and on window functions (b). Only the full model can faithfully synthesize the structured patterns (e.g. the strip textures) in (a) and avoid artifacts and contour distortions in (b).

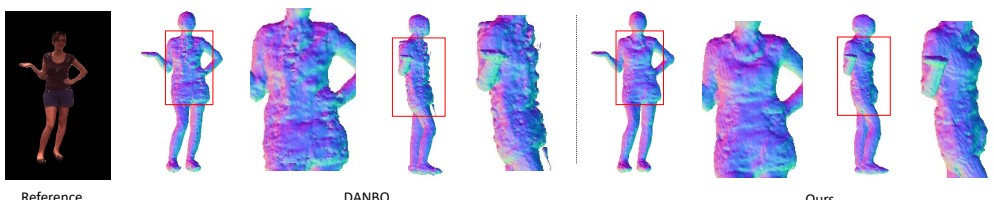

Figure 8: **Geometry reconstruction.** Our method yields more precise, less noisy shape estimates. Some noise remains as no template mesh or other surface prior is used.

### 4.3 NOVEL POSE RENDERING

We follow prior work and measure the quality of novel pose synthesis by training on the first part of a video and testing on the remaining frames. Only the corresponding 3D pose is used as input. This tests the generalization of the learned pose modulation strategy and applicability to animation.

We present the visual comparisons for the Human3.6M dataset in Fig. 6, where our method shows superior results than baselines in terms of fine-grained and consistent renderings. Specifically, our method generates sharper details such as those seen in wrinkles and better texture consistency, as exemplified by the clearer marker (highlighted by boxes). Tab. 2 further quantitatively verifies that our method generalizes well to both held-out poses and out-of-distribution poses across the entire test set. Note that no methods match wrinkle locations perfectly, as these form chaotically, dependent on the past motion, not just on the single frame used by all methods for conditioning. Learning such motion dynamics remains an open problem that is orthogonal to learning pose-dependent detail.

Moreover, we provide the visual comparisons in Fig. 4 and the quantitative metrics in Tab. 1 (a) for the high-resolution outdoor monocular video sequences MonoPerfCap (Xu et al., 2018). Being consistent with the results on the Human3.6M sequences, our method presents better capability in learning a generalized model from monocular videos.

Furthermore, we test the animation ability of our approach by driving models learned from the Human3.6M dataset with extreme out-of-distribution poses from AIST (Li et al., 2021a). The qualitative results shown in Fig. 9 validate that even under extreme pose variation our approach produces plausible body shapes with desired texture details while the baseline show severe artifacts. Here no quantitative evaluations are performed since no ground truth data is available.

Table 2: **Novel-view and novel-pose synthesis results, averaged over the Human3.6M test set.** Our method benefits from the explicit frequency modulations, yielding better perceptual quality, reaching the best overall score in all metrics.

|  | Novel view | | | Novel pose | | |
|---|---|---|---|---|---|---|
|  | PSNR↑ | SSIM↑ | LPIPS↓ | PSNR↑ | SSIM↑ | LPIPS↓ |
| **Template/Scan-based prior** | | | | | | |
| NeuralBody | 23.36 | 0.905 | 0.140 | 22.81 | 0.888 | 0.157 |
| Anim-NeRF | 23.34 | 0.897 | 0.157 | 22.61 | 0.881 | 0.170 |
| ARAH[†] | 24.63 | 0.920 | 0.115 | 23.27 | 0.897 | 0.134 |
| **Template-free** | | | | | | |
| A-NeRF | 24.26 | 0.911 | 0.129 | 23.02 | 0.883 | 0.171 |
| DANBO | 24.69 | 0.917 | 0.116 | 23.74 | 0.901 | 0.131 |
| TAVA | 24.72 | 0.919 | 0.124 | 23.52 | 0.899 | 0.141 |
| **Ours** | **25.06** | **0.921** | **0.110** | **24.15** | **0.906** | **0.124** |

[†]: using public release that differs to Wang et al. (2022).

### 4.4 GEOMETRY VISUALIZATION

In Fig. 8, we analyze the geometry reconstructed with our approach against reconstructions from the baseline. Our method captures better body shapes and per-part geometry. Specifically, our results present overall more complete body outline and a smoother surface. In contrast, the baselines predict more noisy blobs near the body surface. Together with the results of novel view and novel pose

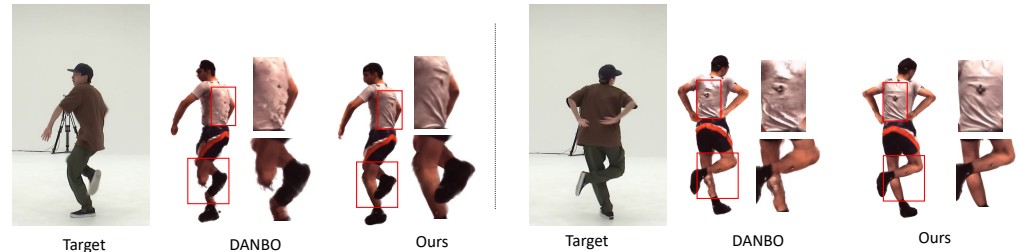

Figure 9: **Animation capability.** Our method maintains the reconstructed high frequency details when retargeting and creates fewer artifacts, lending itself for retargeting.

synthesis, we also attribute our more consistent rendering results across novel views than the baselines to better geometry preservation. This might suggest that more faithful modeling of geometry is also beneficial for the visual fidelity as shown in the appended video.

## 4.5 ABLATION STUDY RESULTS

To test the significance of each component, we conduct the ablation study on all Human3.6M sequences with six ablated models: **1.** Only preserve the upper branch network with GNN features as $\mathrm{onlyGNN}$; **2.** Only preserve the bottom branch network without frequency modulation as $\mathrm{onlySyn}$; **3.** For the two-stage window function, we only preserve $w_i^p$ to evaluate the effectiveness of the spatial similarities between $x$ and all bone parts as only $w_i^p$; **4.** We only preserve $w_i^f$ for the importance of max-pooled object-level feature as only $w_i^f$; **5.** We remove the whole window function as no window. **6.** We concatenate all pose parameters into one vector and replace the GNN layers with one single MLP to evaluate the effectiveness of per-part modeling as $\mathrm{noGNN}$; see Fig. 15 for detailed architecture. In Fig. 7,we present the qualitative ablation study results for S9. Specifically, the $\mathrm{onlyGNN}$ model is prone to produce blurry textures due to its low frequency bias while the $\mathrm{onlySyn}$ model introduces noisy artifacts near stripe textures. Similar to $\mathrm{onlySyn}$, the $\mathrm{noGNN}$ model blurs the stripe patterns with noisy artifacts. Only the combination yields their full advantage and successfully synthesizes structured patterns, as shown in Fig. 7 (a). As mentioned in method section, $w_i^p$ and $w_i^f$ cater for the relationships in position space and feature space, respectively. As shown by the results of novel pose rendering (Fig. 7 (b)), neither using $w_i^p$ or $w_i^f$ alone suffices to produce the image quality of the full model, demonstrating the necessity of all contributions. Tab. 1 (b) lists our corresponding quantitative results of all sequences to further support our statement.

Tab. 1 (a) presents the quantitative scores from the MonoPerfCap dataset, showcasing the performance of both the $\mathrm{onlyGNN}$ and $\mathrm{onlySyn}$ models. Similar to the result differences shown in Tab. 1 (b), our full model consistently outperforms these two ablated models, which reveals our full model's adaptability to diverse in-the-wild scenarios depicted in high-resolution images. With all these results, we conclude that each component clearly contribute to the empirical success of the full model. More ablation studies on the pose-dependent frequency modulation can be found in the appendix.

## 5 CONCLUSION

We introduce a novel, frequency-based framework based on NeRF (Mildenhall et al., 2020) that enables the accurate learning of human body representations from videos. The main contribution of our approach is the explicit integration of desired frequency modeling with pose context. When compared to state-of-the-art algorithms, our method demonstrates improved synthesis quality and enhanced generalization capabilities, particularly when faced with unseen poses and camera views.

**Acknowledgements** This work was supported by the Natural Sciences and Engineering Research Council of Canada (NSERC) Discovery Grant and by Advanced Research Computing at the University of British Columbia. It was also partially supported by the Wallenberg AI, Autonomous Systems and Software Program (WASP) funded by the Knut and Alice Wallenberg Foundation, Sweden. We thank Shih-Yang Su for his help and discussions. We also thank ARC at UBC and Compute Canada for providing computational resources.

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

# Appendices

In this part, we first present the details about method implementation, used dataset and data split. Then we provide more comparison results on the frequency histograms, geometry visualization and motion retargeting. We also attach more ablation studies to emphasize the significance of pose-guided frequency modulation and window functions. More qualitative results for novel view synthesis and novel pose rendering are provided as well. Finally, we discuss the limitations and social impacts of this project. See the attached video for the animation and geometry visualization results.

## A    IMPLEMENTATION DETAILS

For consistency, we maintain the same hyper-parameter settings across various testing experiments, including the loss function with weight $\lambda_s$, the number of training iterations, and the network capacity and learning rate. All the hyper-parameters are chosen depending on the final accuracy on chosen benchmarks. Our method is implemented using PyTorch (Paszke et al., 2019). We utilize the Adam optimizer (Kingma & Ba, 2014) with default parameters $\beta_1 = 0.9$ and $\beta_2 = 0.99$. We employ the step decay schedule to adjust the learning rate, where the initial learning rate is set to $5 \times 10^{-4}$ and we drop the learning rate to $10\%$ every $500000$ iterations. Like former methods (Su et al., 2021; 2022), we set $\lambda_s = 0.001$ and $N_B = 24$ to accurately capture the topology variations and avoid introducing unnecessary training changes. The learnable parameters in GNN, window function and the frequency modulation part are activated by the Sine function while other parameters in the neural field $F$ are activated by Relu (Agarap, 2018). We train our network on a single NVidia RTX 3090 GPU for about 20 hours.

## B    MORE DETAILS ABOUT DATASETS

Follow the experimental settings of previous methods (Su et al., 2021; 2022), we choose the Human3.6M (Ionescu et al., 2011) and MonoPerfCap (Xu et al., 2018) as the evaluation benchmarks. These datasets cover the indoor and outdoor scenes captured by monocular and multi-view videos. Specifically, we use a total of 7 subjects for evaluation under the same evaluation protocol as in AnimNeRF (Peng et al., 2021a). We compute the foreground images with (Gong et al., 2018) to focus on the target characters. Likewise, we adopt the identical pair of sequences and configuration as employed in A-NeRF (Su et al., 2021): Weipeng and Nadia, consisting of 1151 and 1635 images each, with a resolution of $1080 \times 1920$. We estimate the human and camera poses using SPIN (Kolotouros et al., 2019) and following pose refinement (Su et al., 2021). We apply the released DeepLabv3 model (Chen et al., 2017) to estimate the foreground masks. The data split also stays the same as the aforementioned methods for a fair comparison.

## C    MORE RESULTS

**Histogram Comparisons.** Showing the frequency histograms of different frames, like the Fig. 1 in the main text, appear to be a clear solution to demonstrate our motivation. Thus we provide more examples and corresponding analysis here. Using the two close-ups in Fig. 1 of the main text, we present the corresponding frequency histograms for each method in Fig. 10. To test our effectiveness when training over long sequences, we provide the histogram results of two frames collected in the ZJU-MoCap (Peng et al., 2021b) dataset, in Fig. 11. Compared to DANBO, our method can produce more similar contours to the ground truth histograms. Additionally, the matched histogram distances (shown below the histogram subfigures and denoted as **F-Dist**) further confirm our advantages of producing adaptive frequency distributions which is the key point of our method.

Besides measuring the holistic histogram similarities, we directly compute a frequency map by regarding the standard deviation (STD) value at each pixel as a gray-scale value. As evidenced in Fig. 11, our method provides significantly improved results, as represented by the error images between the output frequencies and the ground truth values. All these results reveal that our method can faithfully reconstruct the desired frequency distributions both locally and holistically.

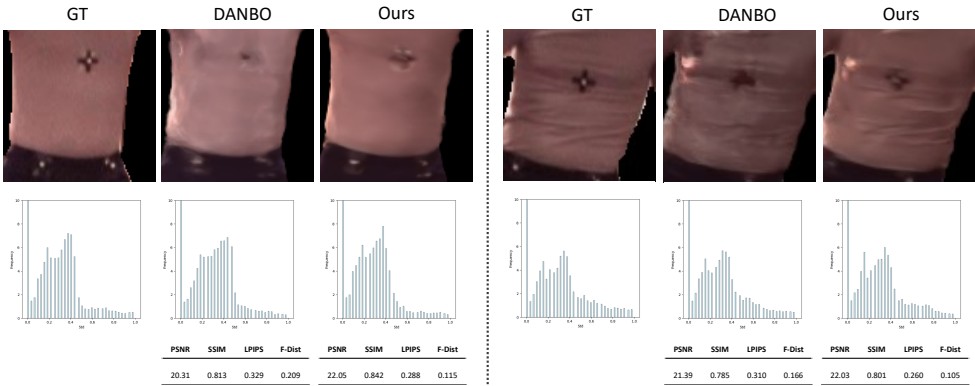

Figure 10: **Motivation Demonstration on Human3.6M frames.** Using the two frames in Fig. 1 of the main text, we present the frequency histograms, compute three image quality metrics (e.g. PSNR, SSIM, LPIPS) and the distances between the output frequency map and ground truth values (F-Dist) to justify our pose-guided frequency modulation. Compared to DANBO which modulates frequencies implicitly, our method can synthesize higher-quality images with more adaptive frequency distributions across different pose contexts.

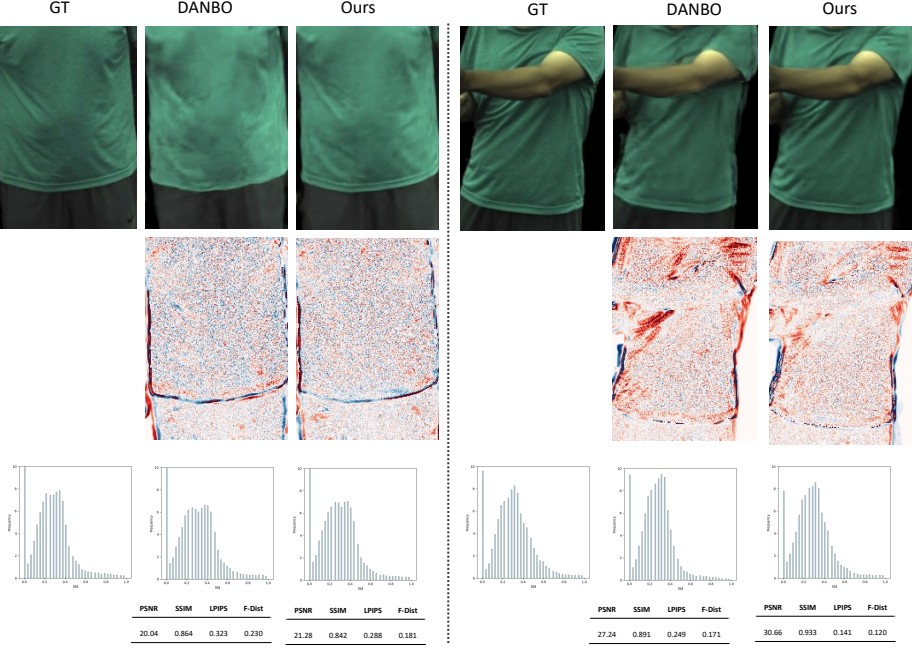

Figure 11: **Motivation Demonstration on ZJU-Mocap frames.** To test our effectiveness over long video sequences, we qualitatively and quantitatively evaluate our method with two ZJU-Mocap frames. Being consistent with the findings in Fig. 10, our method outperforms DANBO with more similar frequency histograms ($3^{rd}$ row) and better quantitative metrics on image quality and frequency modeling (last row). Additionally, we further illustrate the color-coded frequency error maps ($2^{nd}$ row) for both methods to show that our method can faithfully reconstruct the desired frequency distributions both locally and holistically. For the left frame with smooth patterns, our method introduces slightly few frequency errors as it is easy to model low-frequency variations. On the other hand, for the right frame with much more high-frequency wrinkles, our method faithfully reproduces the desired frequencies with significantly less errors, which clearly demonstrate the importance of our pose-guided frequency modulation. Here red denotes positive and blue denotes negative errors.

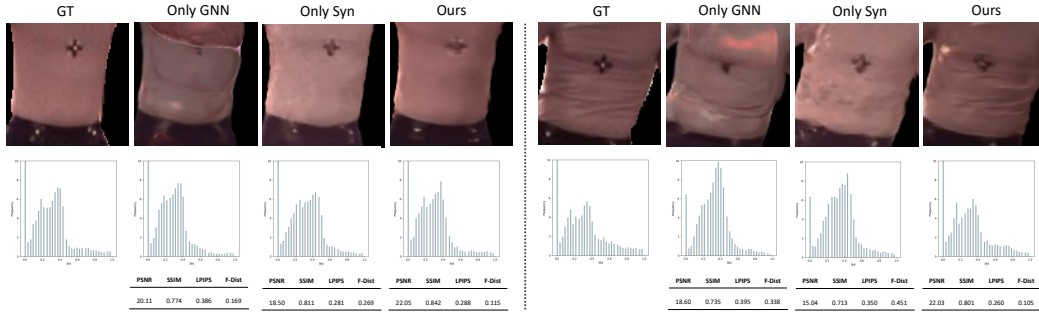

Figure 12: **Abaltion study on the network components with frequency analysis.** Our full model produces more adaptive frequency distributions and higher image quality than the ablated models.

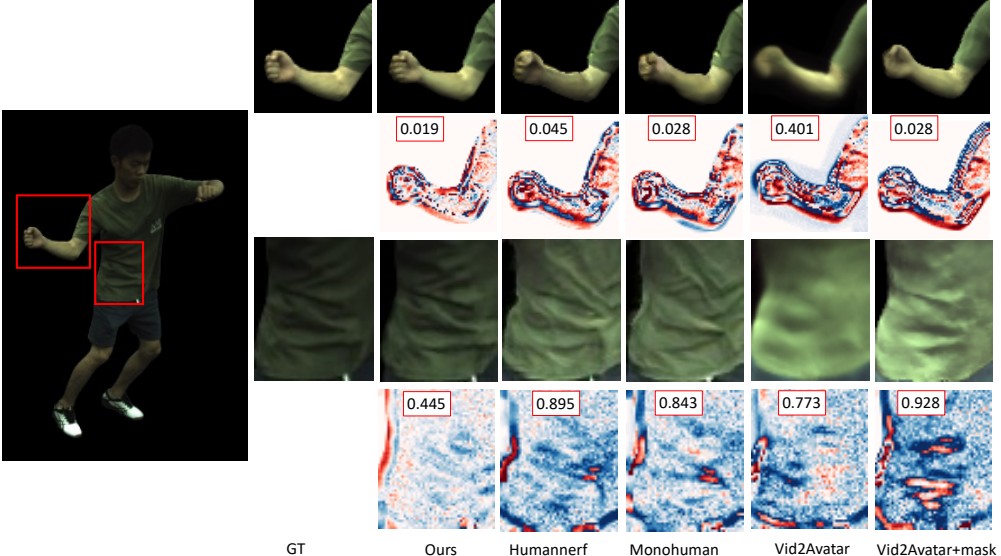

Figure 13: **Visual comparisons on ZJU-Mocap with modern baselines.** Among all methods, only our model can successfully synthesize the avatar fist with accurate arm contours (top row) and faithfully reconstruct the desired wrinkles (bottom row). In contrast, all baselines either distort or blur the fist patterns (top row) and clothing textures (bottom row). Additionally, the state-of-the-art methods fail to capture the cloth colors. Similar to Fig. 11, we illustrate the frequency error maps and compute the distances between the output frequency map and ground truth values (F-Dist). Being consistent with the findings in Fig. 10 and Fig. 11, the pose-guided frequency modulation can better reconstruct the desired frequency distributions both locally and holistically. Thus more adaptive patterns can be reproduced with reduced errors.

Table 3: **Novel-view synthesis results on the ZJU-Mocap Peng et al. (2021b) test set.** The pose-dependent frequency modulation facilitates generalizing to novel views with one single camera input. Here 'Vid2Avatar' indicates the original Vid2Avatar implementation while 'Vid2Avatar+mask' is the model only focusing on the foreground human representations, thus obtaining better metrics than the original version.

| | S377 | | S387 | | S393 | | S394 | | Avg | |
|---|---|---|---|---|---|---|---|---|---|---|
| | PSNR | SSIM | PSNR | SSIM | PSNR | SSIM | PSNR | SSIM | PSNR | SSIM |
| NeuralBody | 28.90 | 0.967 | 27.15 | 0.953 | 28.30 | 0.957 | 28.41 | 0.956 | 28.19 | 0.958 |
| HumanNeRF | 30.21 | 0.975 | 27.95 | 0.963 | 28.38 | 0.961 | 30.00 | 0.963 | 29.14 | 0.965 |
| MonoHuman | 30.23 | 0.976 | **28.20** | 0.962 | 28.40 | 0.962 | 30.02 | 0.963 | 29.21 | 0.966 |
| Vid2Avatar | 29.52 | 0.951 | 28.10 | 0.938 | 27.19 | 0.929 | 28.95 | 0.932 | 28.44 | 0.938 |
| Vid2Avatar+mask | 29.53 | 0.976 | 28.15 | **0.965** | 27.95 | 0.962 | 29.83 | **0.965** | 28.87 | 0.967 |
| **Ours** | **30.72** | **0.979** | **28.20** | **0.965** | **28.91** | **0.964** | **30.36** | **0.965** | **29.55** | **0.968** |

Table 4: **Novel pose rendering results on the ZJU-Mocap Peng et al. (2021b) test set.** Similar to Tab. 3, our method improves the quantitative results for novel pose rendering.

| | S377 | | S387 | | S393 | | S394 | | Avg | |
|---|---|---|---|---|---|---|---|---|---|---|
| | PSNR | SSIM | PSNR | SSIM | PSNR | SSIM | PSNR | SSIM | PSNR | SSIM |
| NeuralBody | 29.18 | 0.969 | 26.43 | 0.951 | 28.15 | 0.956 | 28.14 | 0.956 | 27.98 | 0.958 |
| HumanNeRF | 30.18 | 0.975 | 27.60 | 0.962 | 28.39 | 0.961 | 28.78 | 0.960 | 28.74 | 0.965 |
| MonoHuman | 30.26 | 0.977 | 27.66 | 0.961 | 28.78 | 0.962 | 29.11 | 0.961 | 28.95 | 0.965 |
| Vid2Avatar | 29.86 | 0.954 | 27.42 | 0.938 | 27.20 | 0.929 | 28.55 | 0.932 | 28.26 | 0.938 |
| Vid2Avatar+mask | 29.85 | 0.977 | 27.38 | **0.964** | 28.09 | 0.962 | 29.25 | **0.963** | 28.64 | 0.967 |
| **Ours** | **30.95** | **0.979** | **27.71** | **0.964** | **28.96** | **0.964** | **29.60** | **0.963** | **29.31** | **0.968** |

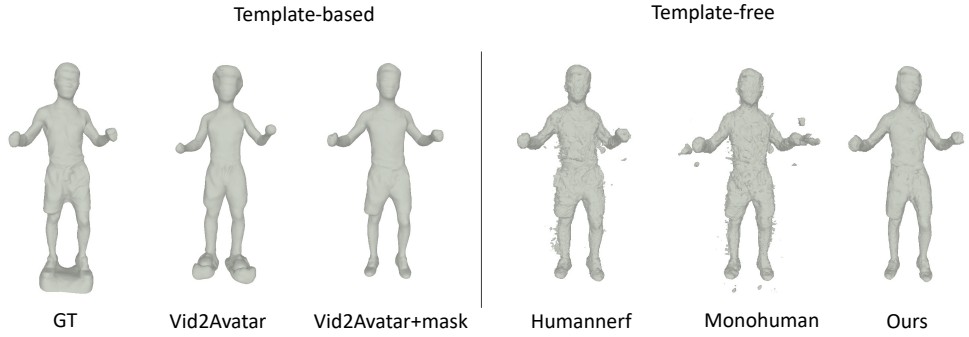

Figure 14: **Geometry reconstruction on ZJU-Mocap S377.** Compared to other template-free methods (HumanNeRF and MonoHuman), our method yields more precise and less noisy body shape. Some bumpy patterns remains as no template mesh or other surface prior is used, in contrast to Vid2Avatar.

**Modern baselines on ZJU-Mocap sequences.** To show our advantages with better generality, we train our method and three baselines, including HumanNeRF Weng et al. (2022), MonoHuman Yu et al. (2023), and Vid2Avatar Guo et al. (2023), over four ZJU-Mocap sequences (S377, S387, S393 and S394) following the experimental settings of HumanNeRF and MonoHuman. All results are achieved through their public codes with default hyper-parameters. Note that, we denote 'Vid2Avatar' as the original Vid2Avatar implementation and correspondingly 'Vid2Avatar+mask' as the model only focusing on the foreground human representations with better empirical results.

Table 5: **Geoemtry Reconstruction on the ZJU-Mocap Peng et al. (2021b) test set.** The pose-dependent frequency modulation improves template-free baselines with more faithful reconstruction and achieves similar results to template-based method.

| | S377 | S387 | S393 | S394 | Avg |
|---|---|---|---|---|---|
| **Template/Scan-based prior** | | | | | |
| Vid2Avatar | 0.1180 | 0.4696 | 0.3330 | 0.2480 | 0.2921 |
| Vid2Avatar+mask | 0.0336 | 0.0283 | 0.0404 | 0.0410 | 0.0359 |
| **Template-free** | | | | | |
| HumanNeRF | 0.0401 | 0.0777 | 0.0609 | 0.0543 | 0.0582 |
| MonoHuman | 0.0897 | 0.0801 | 0.0836 | 0.1006 | 0.0885 |
| **Ours** | 0.0337 | 0.0365 | 0.0455 | 0.0441 | 0.0400 |

Table 6: Ablation studies on full Human3.6M sequences. Our full model improves over all ablated baselines, showing the necessity of all components.

| | onlyGNN | noGNN | onlySyn | $only\ w_i^p$ | $only\ w_i^f$ | $no\ window$ | Ours (full) |
|---|---|---|---|---|---|---|---|
| **Novel view** | | | | | | | |
| PSNR↑ | 24.72 | 24.90 | 24.29 | 24.99 | 23.63 | 23.32 | **25.06** |
| SSIM↑ | 0.914 | 0.918 | 0.911 | 0.920 | 0.835 | 0.842 | **0.921** |
| LPIPS↓ | 0.125 | 0.112 | 0.124 | 0.110 | 0.272 | 0.305 | **0.110** |
| **Novel pose** | | | | | | | |
| PSNR↑ | 23.72 | 23.64 | 23.74 | 24.00 | 22.97 | 22.42 | **24.15** |
| SSIM↑ | 0.898 | 0.900 | 0.900 | 0.905 | 0.824 | 0.832 | **0.906** |
| LPIPS↓ | 0.140 | 0.129 | 0.138 | 0.125 | 0.283 | 0.310 | **0.124** |

(b)

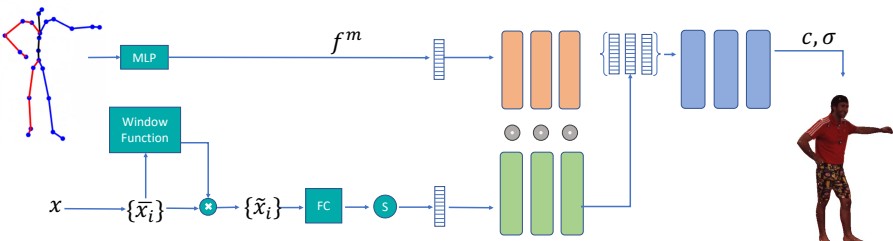

Figure 15: **Architecture of the ablated** noGNN **model.** Compared to our full model, we achieve noGNN by removing graph neural networks and directly feeding the joint parameters as a whole to one single Multi-Layer Perceptron (MLP) for frequency modulation.

In Fig. 13, we illustrate the visual comparisons for our method and chosen baselines. Both Human-NeRF and MonoHuman blur the highlighted avatar fist while we can faithfully reproduce the realistic patterns on the top row. Additionally, on the bottom, the pose-based frequency modulation enables smoother body contours and synthesizes more adaptive wrinkles (e.g. the wrinkle directions) while the baselines all fail to produce accurate cloth colors. The provided error images also underline that our method can more successfully simulate the ground truth images. Similar to Fig. 11, we compute the matched histogram distances (**F-Dist**) for the close-ups. The lowest frequency distance demonstrates that our method can best generate adaptive textures with appropriate frequency distributions. Thus better dynamic wrinkles are enabled. Tab. 3- 4 lists the metric comparisons for novel view synthesis and novel pose rendering, further championing our former observations.

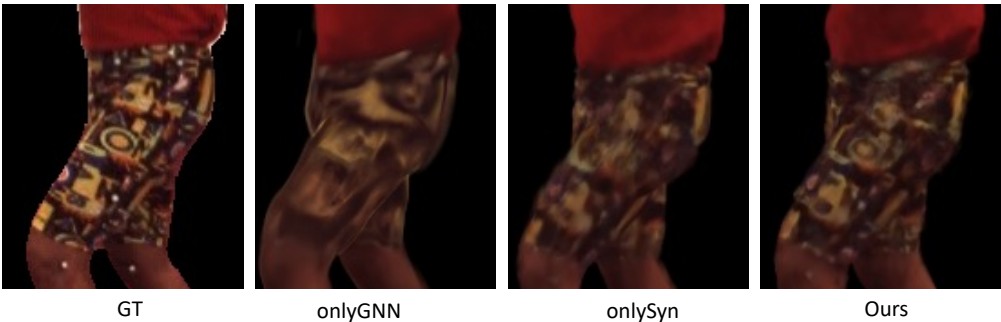

Figure 16: Ablation study on sub-branch networks with novel pose rendering.

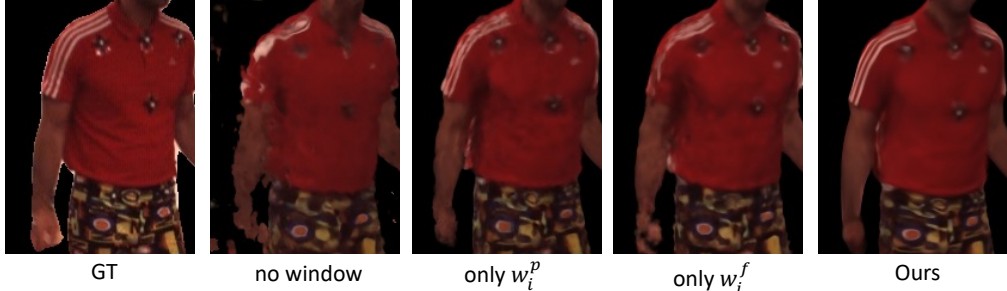

Figure 17: Ablation study on window functions with novel view synthesis.

Like ARAH Wang et al. (2022), we follow its settings and offer the geometry comparison results. Specifically, we also apply NeuS Wang et al. (2021) to obtain the pseudo geometry ground truth and compute quantitative metrics with L2 Chamfer Distance (CD). We show qualitative results in Fig. 14 and quantitative results in Tab. 5. Our method possesses better geometry reconstruction with smoother body surfaces than the template-free methods (HumanNeRF and MonoHuman) which introduce obvious floating artifacts in empty space. The geometry scores are slightly worse than Vid2Avatar which is a template based baseline and relies on SMPL parameters as a prior. The SDF-based surface rendering also helps Vid2Avatar achieve smoother shapes.

**Complete Metrics on Human3.6M sequences.** Besides the overall average numbers in the Tab. 2 of the main text, we also report a per-subject breakdown of the quantitative metrics against all baseline methods. Specifically, Tab. 7 lists the scores for the novel view synthesis while Tab. 8 details each method's results in novel pose rendering. Being consistent with the visual results shown in Fig. 5 and 6 of the main text, our method almost outperforms all baselines for all subjects.

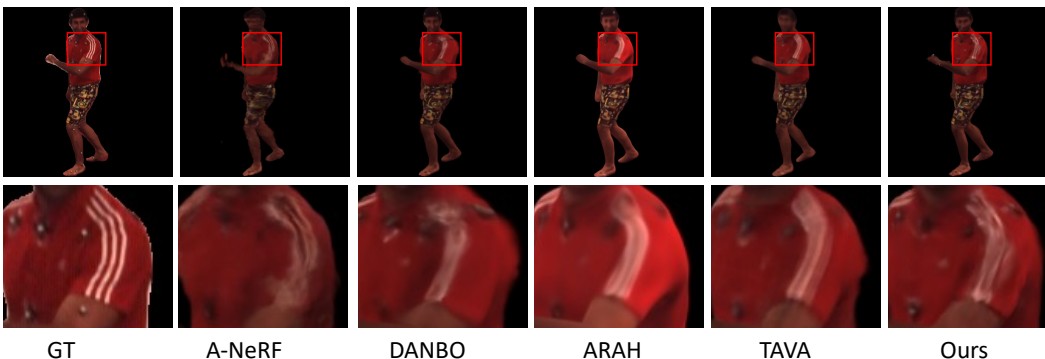

Figure 18: **Failure case.** How to generalize to challenging cases is still an open problem, where all methods fail to capture the stripe-wise textures under this pose.

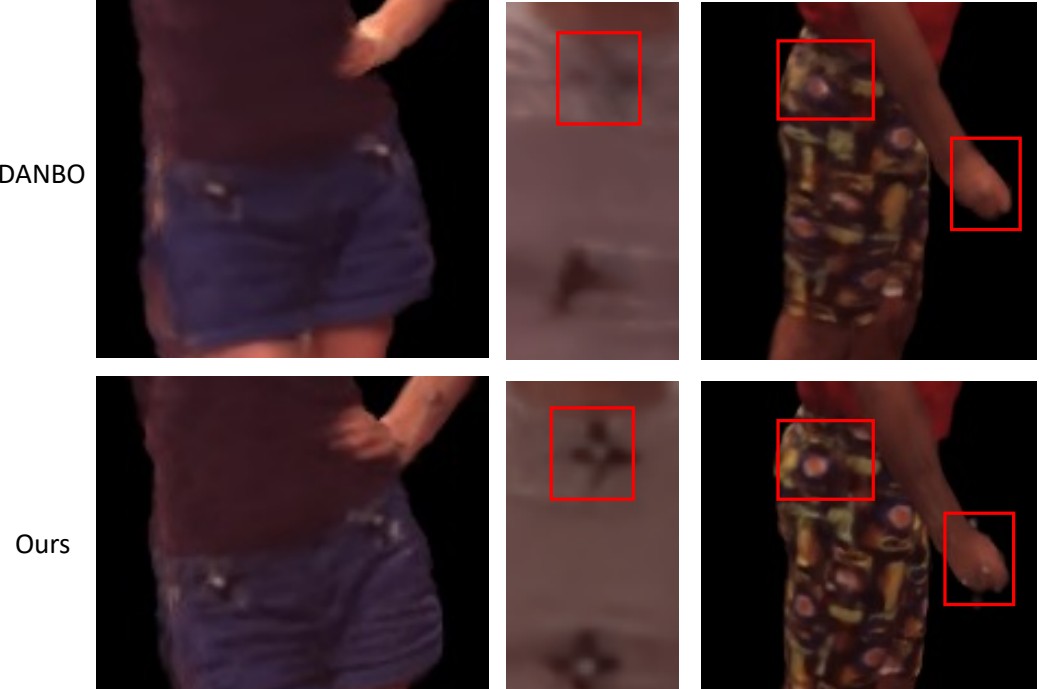

Figure 19: **Additional comparison results with DANBO for novel view synthesis.** Due to the adaptive frequency modulation, our method can better synthesize the shape contour (e.g. hands on $1^{st}$ column), the sharp patterns (e.g. the marker on $2^{nd}$ column), and high-frequency details (e.g. the wrinkles on $1^{st}$ column and the pant textures on $3^{rd}$ column). Otherwise, DANBO which achieves frequency learning implicitly, blurs the sharp patterns and smoothes the fine-grained details.

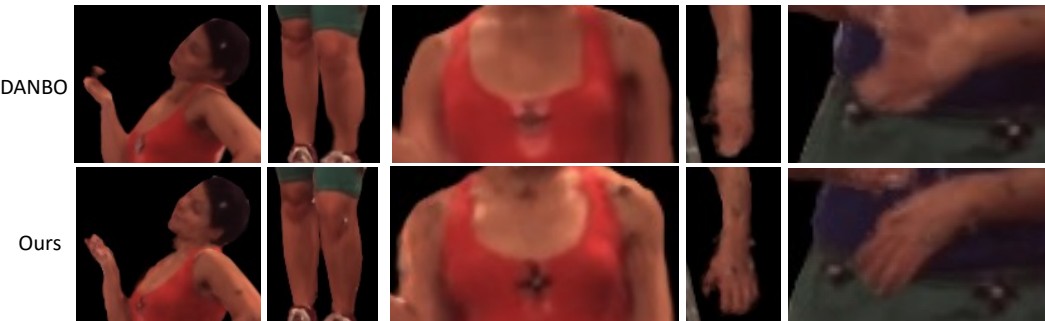

Figure 20: **Additional comparison results with DANBO for novel pose rendering.** Due to the adaptive frequency modulation, our method successfully reduces the noisy artifacts on $1^{st}$ column, reproduces the white markers on $2^{nd}$ column and the black marker on $3^{rd}$ column, reconstructs the sharp shape contours (e.g. the hand) on both $4^{th}$ and $5^{th}$ columns. Otherwise, DANBO which achieves frequency learning implicitly, blurs the sharp contours and smoothes the significant patterns with fine-grained details.

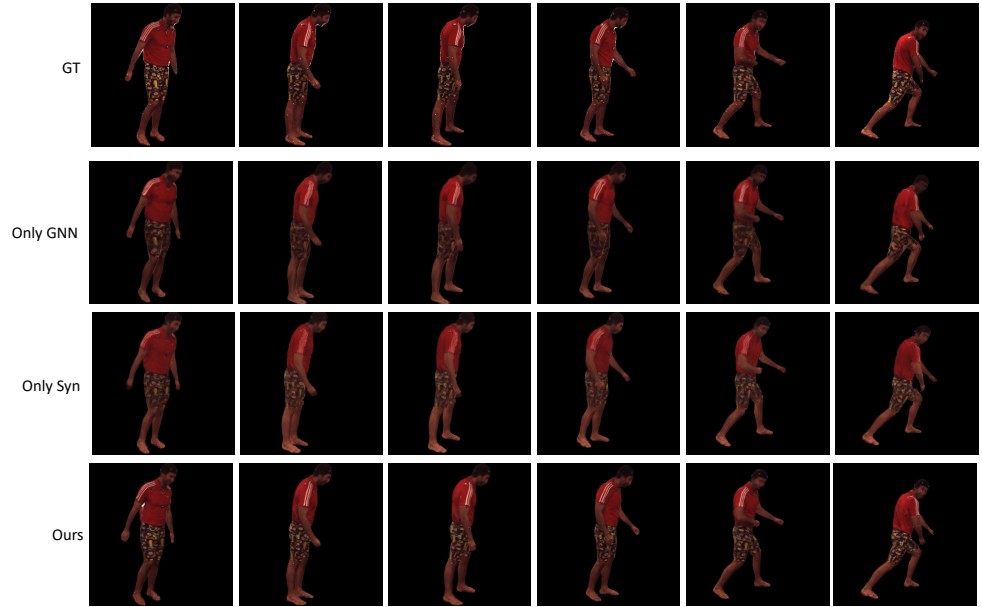

Figure 21: **Abaltion study on the network components with time consistency analysis.** Our full model can consistently produce more adaptive details (e.g. in the pant region), synthesize more structured textures (e.g. the stripes) and preserve more realistic contours (e.g. the leg shape). See texts for details.

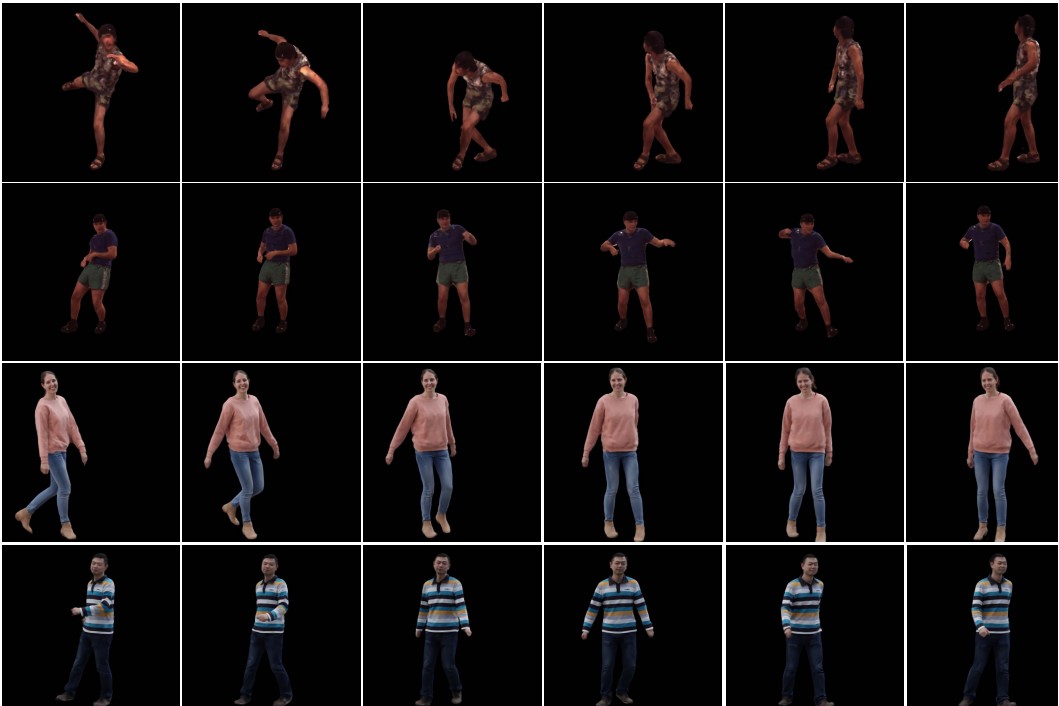

Figure 22: **Unseen pose renderings for the sequences** from both Human3.6M (top two rows) and MonoPerfCap (bottom two rows). Our network is robust to various poses across different datasets.

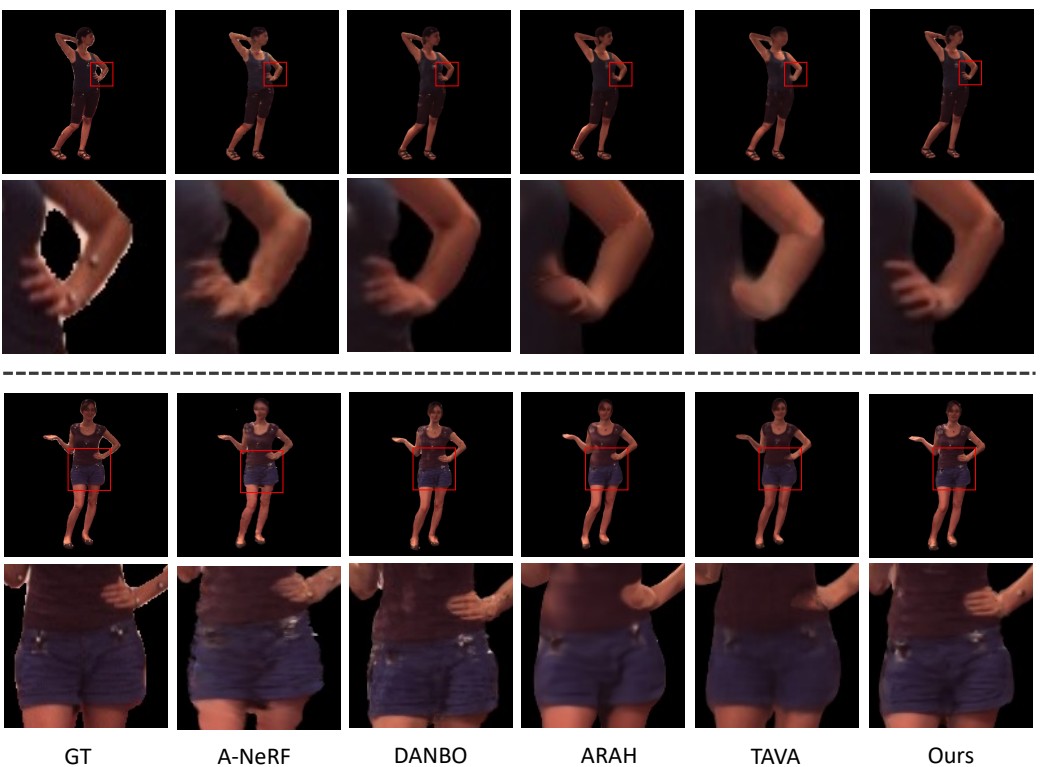

GT          A-NeRF          DANBO          ARAH          TAVA          Ours

Figure 23: **Visual comparisons for novel view synthesis ($1^{st}$ row) and novel pose rendering ($2^{nd}$ row).** Compared to baselines, we can faithfully reconstruct the shape boundaries (e.g. the hands on both $1^{st}$ and $2^{nd}$ rows) and the high-frequency details (e.g. the dynamic wrinkles on $2^{nd}$ row).

Table 7: **Novel-view synthesis results on the Human3.6M Ionescu et al. (2011) test set.** Our method benefits from the explicit frequency modulations, leading to better perceptual quality. It matches or outperforms all baselines across subjects, reaching the best overall score in all three metrics.

| | S1 | | | S5 | | | S6 | | | S7 | | | S8 | | | S9 | | | S11 | | | Avg | | |
|---|---|---|---|---|---|---|---|---|---|---|---|---|---|---|---|---|---|---|---|---|---|---|---|---|
| | PSNR↑ | SSIM↑ | LPIPS↓ | PSNR | SSIM | LPIPS | PSNR | SSIM | LPIPS | PSNR | SSIM | LPIPS | PSNR↑ | SSIM↑ | LPIPS↓ | PSNR | SSIM | LPIPS | PSNR | SSIM | LPIPS | PSNR | SSIM | LPIPS |
| **Template/Scan-based prior** | | | | | | | | | | | | | | | | | | | | | | | | |
| NeuralBody | 22.88 | 0.897 | 0.139 | 24.61 | 0.917 | 0.128 | 22.83 | 0.888 | 0.155 | 23.17 | 0.915 | 0.132 | 21.72 | 0.894 | 0.151 | 24.29 | 0.911 | 0.122 | 23.70 | 0.896 | 0.168 | 23.36 | 0.905 | 0.140 |
| Anim-NeRF | 22.74 | 0.896 | 0.151 | 23.40 | 0.895 | 0.159 | 22.85 | 0.871 | 0.187 | 21.97 | 0.891 | 0.161 | 22.82 | 0.900 | 0.146 | 24.86 | 0.911 | 0.145 | 24.76 | 0.907 | 0.161 | 23.34 | 0.897 | 0.157 |
| ARAH† | 24.53 | 0.921 | 0.103 | 24.67 | 0.921 | 0.115 | 24.37 | 0.904 | 0.133 | 24.41 | 0.922 | 0.115 | **24.15** | **0.924** | **0.104** | 25.43 | 0.924 | 0.112 | 24.76 | 0.918 | 0.128 | 24.63 | 0.920 | 0.115 |
| **Template-free** | | | | | | | | | | | | | | | | | | | | | | | | |
| A-NeRF | 23.93 | 0.912 | 0.118 | 24.67 | 0.919 | 0.114 | 23.78 | 0.887 | 0.147 | 24.40 | 0.917 | 0.125 | 22.70 | 0.907 | 0.130 | 25.58 | 0.916 | 0.126 | 24.38 | 0.905 | 0.152 | 24.26 | 0.911 | 0.129 |
| DANBO | 23.95 | 0.915 | 0.107 | 24.85 | 0.923 | 0.107 | 24.54 | 0.903 | 0.129 | 24.45 | 0.920 | 0.113 | 23.36 | 0.917 | 0.116 | 26.15 | 0.925 | 0.108 | 25.58 | 0.917 | 0.127 | 24.69 | 0.917 | 0.116 |
| TAVA | **25.28** | **0.928** | 0.108 | 24.00 | 0.916 | 0.122 | 23.44 | 0.894 | 0.138 | 24.25 | 0.916 | 0.130 | 23.71 | 0.921 | 0.116 | 26.20 | 0.923 | 0.119 | **26.17** | **0.928** | 0.133 | 24.72 | 0.919 | 0.124 |
| **Ours** | 24.83 | 0.922 | **0.102** | **24.97** | **0.925** | **0.102** | **24.55** | **0.903** | **0.124** | **24.65** | **0.923** | **0.107** | 24.11 | 0.922 | 0.108 | **26.39** | **0.929** | **0.100** | 25.88 | 0.921 | **0.128** | **25.06** | **0.921** | **0.110** |

†: we evaluate using the officially released ARAH, which has undergone refactorization, resulting in slightly different numbers to the ones in Wang et al. (2022).

Table 8: **Novel pose rendering results on the Human3.6M Ionescu et al. (2011) test set.** Our pose guided frequency modulation pipeline generalizes better across unseen poses.

| | S1 | | | S5 | | | S6 | | | S7 | | | S8 | | | S9 | | | S11 | | | Avg | | |
|---|---|---|---|---|---|---|---|---|---|---|---|---|---|---|---|---|---|---|---|---|---|---|---|---|
| | PSNR↑ | SSIM↑ | LPIPS↓ | PSNR | SSIM | LPIPS | PSNR | SSIM | LPIPS | PSNR | SSIM | LPIPS | PSNR↑ | SSIM↑ | LPIPS↓ | PSNR | SSIM | LPIPS | PSNR | SSIM | LPIPS | PSNR | SSIM | LPIPS |
| **Template/Scan-based prior** | | | | | | | | | | | | | | | | | | | | | | | | |
| NeuralBody | 22.10 | 0.878 | 0.143 | 23.52 | 0.897 | 0.144 | 23.42 | 0.892 | 0.146 | 22.59 | 0.893 | 0.163 | 20.94 | 0.876 | 0.172 | 23.05 | 0.885 | 0.150 | 23.72 | 0.884 | 0.179 | 22.81 | 0.888 | 0.157 |
| Anim-NeRF | 21.37 | 0.868 | 0.167 | 22.29 | 0.875 | 0.171 | 22.59 | 0.884 | 0.159 | 22.22 | 0.878 | 0.183 | 21.78 | 0.882 | 0.162 | 23.73 | 0.886 | 0.157 | 23.92 | 0.889 | 0.176 | 22.61 | 0.881 | 0.170 |
| ARAH† | 23.18 | 0.903 | 0.116 | 22.91 | 0.894 | 0.133 | 23.91 | 0.901 | 0.125 | 22.72 | 0.896 | 0.143 | 22.50 | 0.899 | 0.128 | 24.15 | 0.896 | 0.135 | 23.93 | 0.899 | 0.143 | 23.27 | 0.897 | 0.134 |
| **Template-free** | | | | | | | | | | | | | | | | | | | | | | | | |
| A-NeRF | 22.67 | 0.883 | 0.159 | 22.96 | 0.888 | 0.155 | 22.77 | 0.869 | 0.170 | 22.80 | 0.880 | 0.182 | 21.95 | 0.886 | 0.170 | 24.16 | 0.889 | 0.164 | 23.40 | 0.880 | 0.190 | 23.02 | 0.883 | 0.171 |
| DANBO | 23.03 | 0.895 | 0.121 | **23.66** | 0.903 | 0.124 | 24.57 | 0.906 | 0.118 | 23.08 | 0.897 | 0.139 | 22.60 | 0.904 | 0.132 | 24.79 | 0.904 | 0.130 | 24.57 | 0.901 | 0.146 | 23.74 | 0.901 | 0.131 |
| TAVA | **23.83** | **0.908** | 0.120 | 22.89 | 0.898 | 0.135 | 24.54 | 0.906 | 0.122 | 22.33 | 0.882 | 0.163 | 22.50 | 0.906 | 0.130 | 24.80 | 0.901 | 0.138 | **25.22** | **0.913** | 0.145 | 23.52 | 0.899 | 0.141 |
| **Ours** | 23.73 | 0.903 | **0.114** | 23.65 | **0.905** | **0.117** | **24.77** | **0.908** | **0.117** | **23.59** | **0.904** | **0.133** | **23.16** | **0.909** | **0.126** | **25.12** | **0.908** | **0.122** | 25.03 | 0.907 | **0.143** | **24.15** | **0.906** | **0.124** |

†: we evaluate using the officially released ARAH, which has undergone refactorization, resulting in slightly different numbers to the ones in Wang et al. (2022).

**Visual Comparisons with Baselines.** Pose-modulated frequency learning plays a critical role in our method. To demonstrate the importance of this concept, we present more comparisons with DANBO which performs frequency modeling implicitly. In Fig. 19 and Fig. 20, our method is better at preserving large-scale shape contours as well as fine-grained textures with high-frequency details. Besides the results in Fig. 5 and 6 of the main text, we offer two more characters from Human3.6M sequences to evaluate the results on novel view synthesis and novel pose rendering. As shown in Fig. 23, we can successfully reproduce the detailed shape structures (e.g. the hand on $1^{st}$ row) and high-frequency wrinkles (e.g. $2^{nd}$ row). These findings stay consistent with the discussions in main text and the quantitative results in Tab. 7-8.

**Ablation studies.** We formulate our framework from connections between frequencies and pose contexts. To more comprehensively evaluate the effectiveness of our pose-guided frequency modulation concept, we provide one more visual comparison in Fig. 16. It is clear that only our full model successfully synthesizes high frequency patterns, e.g., shown in the pant region.

To additionally showcase the capabilities in reproducing the frequency distributions, we illustrate the frequency histograms for the ground truth images and the network outputs in Fig. 12. Our full model remarkably reduces the gap to the ground truth histograms qualitatively and quantitatively.

Moreover, as shown in Fig. 21, our full model presents much better time consistency than ablated models. Specifically, the full model constantly preserves more adaptive details (e.g. the patterns in the pant region) than the $\mathrm{onlyGNN}$ model and synthesizes more structured stripe-wise patterns than the $\mathrm{onlySyn}$ model. Moreover, the $\mathrm{onlySyn}$ model distorts the leg shape on the last column.

To highlight the empirical importance of the window function $w_i^p$ and $w_i^f$, Fig. 17 depicts qualitative differences between the ablated baselines and our full model. It is clear that using $w_i^p$ or $w_i^f$ alone cannot produce the image quality of the full model, demonstrating the necessity of the window function design. Tab. 6 additionally presents the quantitative comparisons for all ablation models, further supporting the aforementioned discussions.

**Geometry Visualization.** The attached video visualizes two examples for the geometry reconstruction comparison with DANBO. Like the discussions in the main text, we can present overall more complete body outline and a smoother surface than the baseline. Please see video for details.

**Motion Retargeting.** Generality to unseen human poses is critical to a number of down-streamed applications, e.g. Virtual Reality. We provide two examples in the attached video. Although our

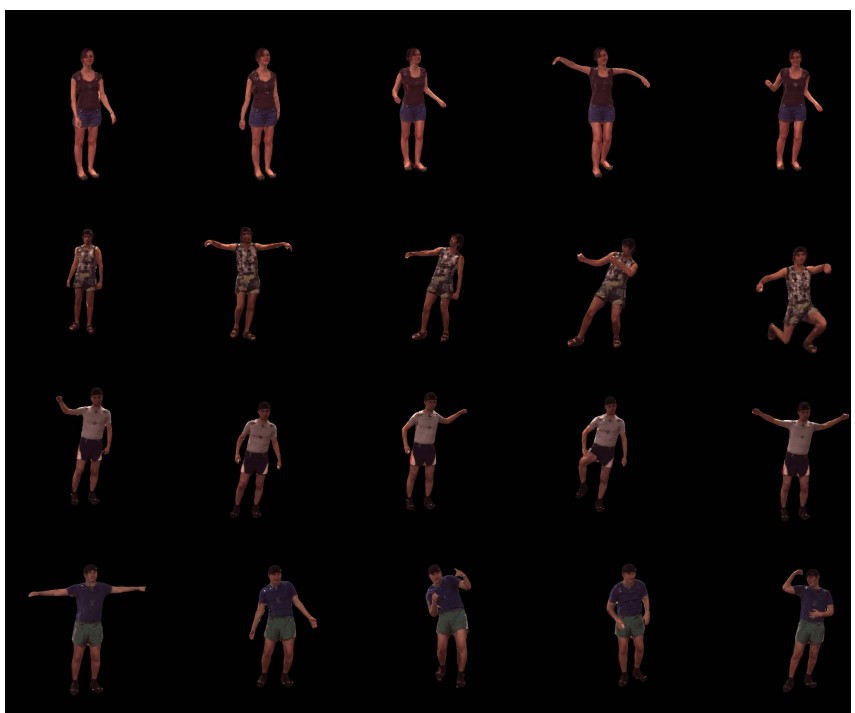

Figure 24: Additional visual results for novel view synthesis. It is clear that our method can faithfully reproduce a larger spectrum of details, from large-scale shape contours (e.g. $1^{st}$ row) to fine-grained textures (e.g. $2^{nd}$ row), across different scenes.

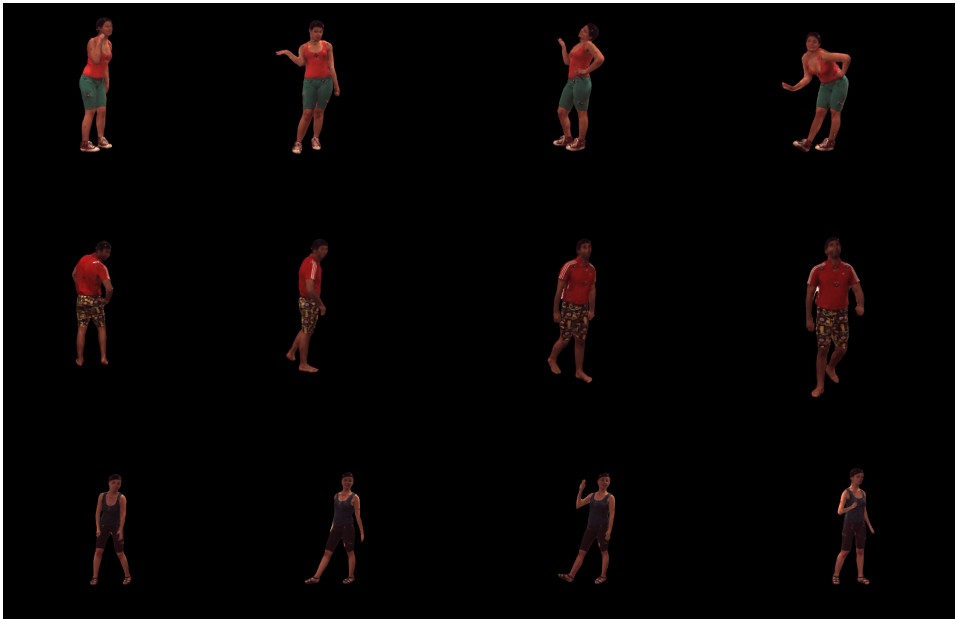

Figure 25: Additional visual results for novel pose rendering. It is clear that our method can generalize well to the unseen poses with different patterns.

model is trained on the Human3.6M sequences, it can consistently be adapted to the unseen poses with challenging movements. The desired time consistency convincingly demonstrates our strong generality to out-of-the-distribution poses.

**More Visual Results.** In order to assess the performance of our method in handling unseen camera views and human skeletons, we provide additional results showcasing novel view synthesis in Fig. 24 and novel pose rendering in Fig. 25. We also illustrate the rendering results for different sequences of Human3.6M and MonoPerfCap datasets for unseen poses in Fig. 22.

From the results, it is evident that our method, with its developed frequency modulation modeling, effectively captures diverse texture details and shape contours even when confronted with human poses that differ significantly from those in the training set. This empirical advantage can be attributed to the adaptive detail modeling capabilities facilitated by pose-modulated frequency learning strategy.

## D  LIMITATIONS AND DISCUSSIONS

Although our method is faster than other neural field approaches, computation time remains a constraint for real-time use. Our method is also person-specific, demanding individual training for each person. And our method heavily relies on accurate camera parameters and lacks support for property editing like pattern transfer. Thus this approach shines with ample training time and available data. Additionally, as shown in Fig. 18, under extreme challenges, our method cannot vividly reproduce the desired patterns but introduces blurry artifacts. However, we would like to note that, how to advance the generalization to such cases is still open since the existing methods suffer from similar or worse artefacts as well.

**Social Impacts.** Our research holds the promise of greatly improving the efficiency of human avatar modeling pipelines, promoting inclusivity for underrepresented individuals and activities in supervised datasets. However, it's imperative to address the ethical aspects and potential risks of creating 3D models without consent. Users must rely on datasets specifically collected for motion capture algorithm development, respecting proper consent and ethical considerations. Furthermore, in the final version, all identifiable faces will be blurred for anonymity.

