# OpenReview forum: "Pose Modulated Avatars from Video"
_ICLR.cc/2024/Conference — ICLR 2024 poster_

### Official Review · Reviewer_PkgA · 2023-10-31

**Soundness:** 3 good
**Presentation:** 3 good
**Contribution:** 2 fair
**Rating:** 6
**Confidence:** 4

**Summary:**

The paper presents a framework that learns an animatable human body model from video without relying on surface information. The approach builds on previous research by incorporating frequency modeling with pose context. The experiments conducted show that the proposed method is capable of generating higher-quality images and has improved generalization abilities when dealing with new poses and viewpoints.

**Strengths:**

* The paper is well-written. Technical details are also well-elaborated.
* Based on the evaluation, the overall quality of the results seems to be satisfactory. Additionally, the quantitative results show better performance compared to DANBO.

**Weaknesses:**

Motivation:
* I find the motivation in Figure 1 to be unclear. The two poses are quite different - the first contains wrinkles while the second doesn't - but their frequency distribution appears quite similar. Perhaps it would be better to choose a sample with poses that are closer to each other and have more significant differences in frequency.
* The paper mentioned that even when a subject is in a similar pose, the frequency distributions can still be distinct. This seems contradictory to the motivation of pose-dependent frequency modulation, as one might wonder why pose-dependent frequency modulation would be beneficial.

Evaluations:
* Are the point locations clearly separated by the learned weights for each part, or are the parts mixed around the joints? It would be best to visualize the weights for the areas where the bones overlap.
* The novel pose results on MonoPerfCap do not match the numbers reported in DANBO. Is there a setup difference?
* Table 1 (b) caption seems to be incorrect. Is the ablation study tested on Human3.6M S9, or is it on MonoPerfCap?
* The ablation should be performed on the complete setup to allow for better comparison with DANBO. Additionally, the current ablation study is unable to determine which specific design element is responsible for the model's improved performance compared to DANBO.
* Is there a specific reason why NeuMan is not being compared with Template/Scan-based prior methods and HumanNeRF for the template-free method in Table 2? A discussion or quantitative comparison would be better.
* It would be more convincing to evaluate the performance of the model on the entire ZJU-Mocap dataset rather than a few selected frames.

Minor:
* The code link seems to be missing.

**Questions:**

Please see the weakness section for details.

---

> ### Author Response · Authors · 2023-11-21
> **Response to Reviewer PkgA (part 1)**
>
> We sincerely thank the reviewer for reading our paper and giving us thoughtful feedback. We carefully respond to each of the summarized concerns and questions. All these feedbacks will be incorporated into our revised manuscript.
>
> **Q1: In Figure 1, the frequency distributions appear to be quite similar.**
>
> We respectfully clarify that these two frequency distributions are actually very different. In Figure 1, we can see that the avatar on the bottom exhibits much more small STD values than the top one: 52 (bottom) vs 23 (top). In Figure 10, we provide a more detailed frequency illustration. For the number of STD values larger than 0.4, the top avatar in Figure 1 contributes much more than the one on the bottom. In contrast, the top avatar in Figure 1 provides much less STD values smaller than 0.4.
>
> **Q2: When a subject is in a similar pose, the frequency distributions can still be distinct.**
>
> As mentioned in the abstract, we hypothesize that different poses necessitate unique frequency assignments. And neglecting this distinction yields noisy artifacts in smooth areas or blurs fine-grained texture and shape details in sharp regions. So we agree with the reviewer that when a subject is in a similar pose, the frequency distributions should be close as well. If there are some misunderstandings / errors in our main text, please let us know.
>
> **Q3: Are the point locations clearly separated by the learned weights for each part?**
>
> As the per-part weights {$w_i$} are a continuous feature vector, instead of being binary. So when two bones overlap, the parts mixed around the joints as the query points are influenced by different bones.
>
> **Q4: Table 1 (b) caption seems to be incorrect.**
>
> Thank you for pointing this out. The ablation studies are performed on Human3.6M S9. We will correct it in the revision.
>
> **Q5: Is there a setup difference for the MonoPerfCap results compared to DANBO?**
>
> Yes, we use tighter masks to highlight the foreground differences.
>
> **Q6: The ablation should be performed on the complete setup.**
>
> Thank you for your suggestions. We follow DANBO’s settings to perform the ablation studies on Human3.6M S9. To verify our motivations comprehensively, we train our **OnlySyn**, **noGNN**, **OnlySyn** ablation models over the whole Human3.6M sequences and report the scores below. Being consistent with the conclusions in the main text, each component does not suffice to provide the optimal performance. Due to the constraints of response phase and computation resources, we will provide other ablation study results over all Human3.6M sequences in the future revision.
>
> |            | onlyGNN | noGNN | onlySyn |  Ours |
> |:----------:|:-------:|:-------:|:-------:|:----:|
> | Novel view |         |         |              |              |
> |    PSNR    |   24.72      |   24.90      |      24.29        |      **25.06**        |
> |    SSIM    |    0.914     |    0.918     |       0.911       |      **0.921**       |
> |    LPIPS   |    0.125     |    0.112      |     0.124         |     **0.110**        |
> | Novel Pose |         |         |              |              |
> |    PSNR    |   23.72      |   23.64      |     23.74         |      **24.15**        |
> |    SSIM    |    0.898     |    0.900     |      0.900        |       **0.906**       |
> |    LPIPS   |    0.140     |    0.129     |      0.138       |      **0.124**       |

---

> > ### Author Response · Authors · 2023-11-21
> > **Response to Reviewer PkgA (part 2)**
> >
> > **Q7: The current ablation study is unable to determine which specific design is responsible for improvements over DANBO.**
> >
> > Please note that the usage of GNNs for separate part-level feature space and the feature aggregation modules are claimed to be the key contributions of DANBO. Conceptually, DANBO is mostly close to our OnlyGNN ablated model which consists of GNN and the learnable window function to aggregate part-level features. According to Table 1 and Table 3-4, we can see that **OnlyGNN** is consistently inferior to DANBO across three metrics as it does not explicitly model part-level feature spaces and depends on simplified feature aggregations. However, with the frequency modulations, our full model significantly outperforms DANBO, especially for novel pose rendering which is more challenging. Thus, we attribute the improved performance over DANBO to the proposed frequency modulations, which is our key contribution.
> >
> > **Q8: Comparisons over ZJU-Mocap sequences.**
> >
> > Thank you for your suggestions. In the common response part, we have attached the comparison results on ZJU-Mocap over HumanNeRF, MonoHuman and Vid2Avatar. We can see that our pose modulated frequency modeling demonstrates superior quantitative advantages compared to all baselines.
> >
> > **Q9: Why not compare with NeuMan.**
> >
> > Thank you for pointing this out. During the response phase, we cannot provide the comparison results to NeuMan, due to the constraints of response phase and computation resources. However, we have already presented our advantages over HumanNeRF, MonoHuman and Vid2Avatar, which compare to it and are known to be stronger baselines (Guo et al., 2023).
> >
> > **We will endeavor to address these points thoroughly in our revised submission.**

---

> ### Comment · Reviewer_PkgA · 2023-11-22
>
> I appreciate the authors for addressing all my concerns through clear explanations and additional experiments within the limited rebuttal time. The idea of using frequency modulation to model wrinkles is interesting, and the results seem convincing, so I'm willing to raise my rating.

---

> > ### Author Response · Authors · 2023-11-22
> > **Thanks for the comments by Reviewer PkgA**
> >
> > Dear Reviewer PkgA,
> >
> > We thank you for your positive response, appreciation of the rebuttal and increase of the score. We especially appreciate that the reviewer recognizes the importance of our motivation and the significance of our experimental results. We value your comprehensive suggestions very much!
> >
> > Best,
> > The Authors

---

### Official Review · Reviewer_SAAP · 2023-11-03

**Soundness:** 2 fair
**Presentation:** 3 good
**Contribution:** 2 fair
**Rating:** 6
**Confidence:** 5

**Summary:**

The authors tackle the problem of human avatar modeling from a monocular video sequence. The paper proposes a pose-driven frequency modulation approach for the underlying NeRF model to achieve a higher accuracy of the rendered images. This approach is shown to be effective for modeling people with tight closing compared to some of the baseline methods.

**Strengths:**

- The problem of avatar modeling has high practical significance
- The frequency modulation approach makes sense to introduce the details in cases where they are required, which can serve as a regularization measure
- The paper is fairly well written

**Weaknesses:**

- The comparison lacks modern baselines, such as Vid2Avatar, MonoHuman, and HumanNeRF, which were referenced in the related work.
- Video results have very low FPS, and therefore, the temporal smoothness of the proposed approach cannot be evaluated.
- It is unclear whether or not GNNs are actually needed for this task, ex. Vid2Avatar uses pose conditioning without GNNs to directly produce the embeddings via an MLP
- No experiments on loose clothing where the method's effectiveness for high-frequency clothing modeling can be asserted.

**Questions:**

- Please address my concerns in the weaknesses section

---

> ### Author Response · Authors · 2023-11-21
> **Response to Reviewer SAAP (part 1)**
>
> We sincerely thank the reviewer for reading our paper and giving us thoughtful feedback. We carefully respond to each of the summarized concerns and questions. All these feedbacks will be incorporated into our revised manuscript.
>
> **Q1: It is unclear whether or not GNNs are actually needed for this task.**
>
> As an example,  Vid2Avatar takes the whole pose information as a condition, but it is only applied for novel view synthesis, not for animation with novel driving poses. GNNs enable learning separate feature spaces for different parts, which was shown to improve disentanglement for animation (Su et al., 2022). Plus the sparse nature of our learnable window functions, the output feature of one query point only attends to a few joints. Thus a more accurate animation with out-of-distribution poses can be captured than directly using the whole skeletons .
>
> To further highlight the importance of GNN, we report the novel view synthesis && novel pose rendering results of an ablation model by removing graph neural networks but directly feeding the joint parameters to one single Multi-Layer Perceptron (MLP) with comparable trainable parameters to our full model. The detailed architecture  of the GNN ablation model is illustrated in Figure 15 of the revised paper. Similar to Vid2Avatar, processing the whole joints together disenables modeling part-level feature space. We train the ablation model over all Human3.6M sequences. As listed in the table below, our full model (the **right** value in each entry) outperforms the ablated model across all metrics. Especially, being consistent with the aforementioned discussions, the usage of GNN enables us to better generalize to novel poses, proving the effectiveness of GNNs and part-based feature learning. Together with other ablation study results (e.g. no window) over full sequences, these quantitative results will be included in future revision.
>
> |       |       S1      |       S5      |       S6      |       S7      |       S8      |       S9      |      S11      |      Avg      |
> |:-----:|:-------------:|:-------------:|:-------------:|:-------------:|:-------------:|:-------------:|:-------------:|:-------------:|
> |  **Novel View** |   |   |   |   |   |   |   |   |
> |  PSNR |  24.70 / **24.83** |  24.88 / **24.97**  |  24.50 / **24.55** |  24.46 / **24.65** |  23.91 / **24.11** |  26.08 / **26.39** |  25.8 / **25.88** |  24.90 / **25.06** |
> |  SSIM | 0.92 / **0.922** | 0.923 /  **0.925** | 0.902 / **0.903** | 0.919 / **0.923** | 0.921 / **0.922** | 0.925 / **0.929** | 0.920 / **0.921** | 0.918 / **0.921** |
> | LPIPS | **0.101** /  0.102 | 0.104 /  **0.102** | 0.125 / **0.124** | 0.112 / **0.107** | 0.110 / **0.108** | 0.105 / **0.100** | **0.125** / 0.128 | 0.112 / **0.110** |
> |  **Novel Pose** |   |   |   |   |   |   |   |   |
> |  PSNR |  23.36 / **23.73** |  23.31 / **23.65** |  24.19 / **24.77** |  22.96 / **23.59** |  22.58 / **23.16** |  24.45 / **25.12** |  24.69 / **25.03** |  23.64 / **24.15** |
> |  SSIM | 0.898 / **0.903** | 0.903 / **0.905** | 0.900 / **0.908** | 0.897 / **0.904** | 0.905 / **0.909** | 0.899 / **0.908** | 0.903 / **0.907** | 0.900 / **0.906** |
> | LPIPS | 0.120 / **0.114** | 0.118 / **0.117** | 0.122 / **0.117** | 0.139 / **0.133** | 0.128 / **0.126** | 0.131 / **0.122** | 0.146 / **0.143** | 0.129 / **0.124** |

---

> ### Author Response · Authors · 2023-11-21
> **Response to Reviewer SAAP (part 2)**
>
> **Q2: Video results have very low FPS.**
>
> In the comparison sequences, for ARAH, we utilized the generated images provided by its authors. These video frames were sampled from the entire sequence at a fixed frame rate (extracting one frame per 30 frames), which is a common strategy. From the video, we can observe that the frame sequences obtained through sampling have adequately approximated the complete input videos.
>
> To further validate the advantages of our method in terms of temporal smoothness, we present the results of DANBO, TAVA and ours on the complete image sequences. In the newly uploaded video, we insert the full frame comparison clip into a section named **Temporal Smoothness Comparisons** after the section **Comparisons for Animation Sequences**. As shown in the attached video, our method demonstrates superior temporal smoothness with consistently adaptive textures.
>
> In addition to the video clips used for comparison (named as **Comparisons for Animation Sequence** in the video), we utilized the full image sets to synthesize the subsequent video results (starting from **More Results for Animation Sequence**). We can observe that our method constantly showcases structured features (e.g., the stripe patterns on the shoulder area) and smooth body contours. Please refer to the results under the video sections titled **More Results for Animation Sequences** and **Motion Retargeting**.
>
> Our technical contribution is also evident in the ablation study section. In comparison to the **OnlySyn** ablation model, our full framework can synthesize a more complete shape contour (e.g., hands) and produce realistic marker textures without artifacts.
>
> Due to the constraints of rebuttal phase and computation resources, the results of ARAH on the complete sequence will be included in the future revision stage.
>
>
> **Q3: No experiments on loose clothing.**
>
> Our goal is to verify the effectiveness of pose modulated frequency learning.
> When the deformed avatar wears soft clothes (e.g. T-shirt), our method can successfully reconstruct the desired wrinkle textures; please refer to Figure 1 and Figure 11 of the main text as an example. Although we have already presented our advantages across different popular datasets, how to extend our method to more challenging datasets with loose clothing is an interesting direction and lies in our future work. Please note that loose clothing calls for modeling dynamics physically, e.g. swinging, which is orthogonal to our current work.
>
> **May we ask if our explanation satisfies you?**

---

> > ### Comment · Reviewer_SAAP · 2023-11-22
> > **Thank you for the rebuttal**
> >
> > I thank the authors for the rebuttal and for trying to address my concerns.
> >
> > The provided quantitative and qualitative results, unfortunately, do not show substantial improvement in the proposed method, and in many cases, it is hard to tell the difference between it and the baselines.
> >
> > In the ablation study, the measured difference has an order of 10^-3 for SSIM/LPIPS and 10^-1 for PSNR, which is insufficient given the sample size of 7 videos to discern whether or not it is statistically significant. The same goes for the comparison with other baselines. Also, given that the numbers are so close, it would be helpful to see qualitative results, too.
> >
> > One question that is lifted from my side by the rebuttal is temporal smoothness. The method does exhibit it, thank you for providing the results.

---

> > > ### Author Response · Authors · 2023-11-23
> > > **Are there further questions before the end of author-reviewer discussion period?**
> > >
> > > We thank Reviewer SAAP again for your last feedback. Currently, both Reviewer McVe and Reviewer PkgA change to be positive and are satisfied with **the importance of our motivation and the significance of our experimental results**, which greatly overlaps with your core questions.
> > >
> > > As the author-reviewer discussion period will end in several hours, we would like to see if our response addressed your concerns regarding the experimental results. Please let us know whether you have any further questions, and we will do our best to reply before the end of the discussion period.

---

> ### Author Response · Authors · 2023-11-22
> **Thank Reviewer SAAP for the comments**
>
> We thank Reviewer SAAP for the feedback. We are glad to see that the temporal smoothness issue has already been addressed. In the following, we will summarize your concerns and our corresponding response point-by-point.
>
> **Q1: The provided quantitative results do not show substantial improvement.**
>
> As mentioned in the **Common Response to All Reviewers** and our response to Q2 of **Response to Reviewer McVe**, our empirical improvement can be **recognized to be significant** according to the most recent literature. For example, in Table 2 of Vid2Avatar, it improves over HumanNeRF by **0.001 in SSIM and 0.3 in PSNR**. Similarly, in the Table 4 of MonoHuman, the full model improves over the baselines by **<0.001 in SSIM and <0.2 in PSNR**. And in the Table 4 of DANBO, the full model improves the ‘Softmax-OOB` baseline with **0.002 in SSIM and 0.38 in PSNR**. Please check their papers for details. Correspondingly, for the provided comparisons over ZJU-Mocap dataset, we improve the best baseline (MonoHuman) with **0.002 in SSIM and 0.34 in PSNR**. Thus, in terms of reported scores, our method has already achieved the typical improvements over prior work. As mentioned in the past response, we also list the comparisons for model complexities and geometry reconstruction. In Table 4 of the revised paper, our method has the about **45.5%** and **121.3%** relative improvements over HumanNeRF and MonoHuman respectively. We incorporate the metrics of rendering results, model complexities and geometry reconstructions into one single table below for more convenient comparisons.
>
> |                       | Vid2Avatar | HumanNeRF | MonoHuman |    Ours    |
> |:---------------------:|:----------:|:---------:|:---------:|:----------:|
> |      PSNR &uarr;      |    28.44   |   29.14   |   29.21   |  **29.55** |
> |      SSIM &uarr;      |    0.938   |   0.965   |   0.966   |  **0.968** |
> | Model Size (M) &darr; |      **0.9**      |     60    |     74    |      3     |
> |    Geometry &darr;    |   0.2921   |   0.0582  |   0.0885  | **0.0400** |
>
>
> We emphasize that our method outperforms the baselines across almost all metrics. PSNR and SSIM metrics are known to differ in magnitude from dataset to dataset, but relative improvements remain very consistent and quantify the overall improvement well, even if small in absolute numbers. Thus we respectfully disagree with the reviewer that the improvement is not statistically substantial (c.f. data size discussion below).
>
>
> **Q2:  The provided qualitative results do not show substantial improvement.**
>
> Following the conventions of former neural avatar papers, we illustrate to exemplify our conceptual advantages. Specifically, the pose guided frequency modulation can facilitate synthesizing adaptive patterns with reduced artifacts. Specifically, in Figure 7 (a), the **noGNN** ablation fails to synthesize the structured stripe with blurry artifacts. For evaluations on ZJU-Mocap, **only** our method can successfully synthesize the avatar fist with accurate arm contours and capture the cloth colors in Figure 13. In contrast, all baselines either distort or blur the fist patterns (top row) and clothing textures with incorrect cloth colors (bottom row). To better illustrate our advantages, we present the frequency error maps which are introduced in Figure 11. It is clear that the pose guided frequency modulation can remarkably reduce the unwanted errors than all baselines. The computed **F-Dist** scores further highlight this point: our method has relatively **47.4%** (top row) and **73.7%** (bottom row) improvements over the best baseline.
>
> For the authors, both qualitative and quantitative results **clearly** demonstrate the effectiveness of pose guided frequency modulation. But we are also happy to further discuss with detailed comments or arguments.
>
> **Q3: Sequence number for ablation studies.**
>
> Following the settings of former baselines, we think we have used sufficient data to achieve a fair comparison and conclusion. For example, DANBO uses **one** Human3.6M sequence (S9) as described in Section 4.4. Both HumanNeRF and MonoHuman apply **six** ZJU-Mocap sequences for ablations as shown in Table 3 and Table 1-2 of their papers respectively. And in our paper, we have used **seven** sequences. Overall, we have made use of 7 Human3.6M sequences, 2 MonoPerfCap sequences and 4 ZJU-Mocap sequences to pinpoint the effectiveness of the proposed framework, each containing dozens to hundreds of frames. It can be seen to be a very **thorough** experimental setting and prior work validated that the magnitude of improvements we attain  is a consistent and significant improvement.

---

### Official Review · Reviewer_McVe · 2023-11-06

**Soundness:** 3 good
**Presentation:** 4 excellent
**Contribution:** 2 fair
**Rating:** 6
**Confidence:** 4

**Summary:**

This paper introduces a method for pose-modulated animatable human avatar creation using Nerfs. The key idea that the paper introduces is to modulate the frequencies of the sinusoidal embeddings used to encode input 3D locations, before they are input into the Nerf's MLP for inference, depending on their local position and nearest bones' pose. These are key to modeling fine-grained texture details of folds caused by bone deformation on clothing, etc. To achieve this the authors propose a module to that first encodes the poses of joints via a graph neural network structure and then predict the frequency modulation of a 3D location based on its local bone's encoded GNN features and position in 2D space relative to the skeletal structure. The authors compare the proposed method to several existing competing methods, both qualitatively and quantitatively, and observe superior performance for their method.

**Strengths:**

In terms of novelty, the problem of animatable neural human avatar creation is a widely studied one. However this paper proposes the original new idea of modulating the frequency bands used in Nerfs to correctly learn to model wrinkles on clothes based on the deforming pose. This idea is conceptually sound and provides an interesting novel insight to the problem of human avatar creation. Modeling the deformation of loose clothing is still a fairly unsolved problem within this domain and hence advances the field forward.

The proposed solution and experimental methodology are technically sound. The paper will well-written and structured. Many details are described in the supplement. The authors have promised to released the code.

**Weaknesses:**

The main weaknesses are in terms of the results and experiments.

1. Overall the results in the supplementary videos are quite blurry. The effect of the improvement in texture quality of the wrinkles with the proposed method are also subtle and hard to really appreciate. The numerical results in Table 2 of the paper correlate with this fact and show marginal numerical improvement in the reported metrics. Do the authors believe these numerical improvements are statistically significant?

2. For the novel view synthesis task, I am curious as to why the authors did not compared against the following several more recent state-of-the-art methods, which result in higher rendering quality.

a) Guo et al., Vid2Avatar: 3D Avatar Reconstruction from Videos in the Wild via Self-supervised Scene Decomposition, CVPR 2023.

b) Jiang et al.,  Neuman: Neural human radiance field from a single video, ECCV 2022.

c) Weng et al., Humannerf: Free-viewpoint rendering of moving people from monocular video, CVPR 2022.

d) Yu et al., MonoHuman: Animatable Human Neural Field from Monocular Video, CVPR 2023.

3. Related also to question 2, is why did the authors choose to not report quantitative metrics for geomtric reconstruction quality and compare it to the existing state-of-the-art methods listed in question 2?

**Questions:**

I would like to see the authors' response to the questions I have posed in the "weaknesses" section of my review.

Overall, I feel that while the idea of pose-conditioned frequency modulation to model wrinkles on clothing is interesting and worth sharing with the wider research community, the rendering quality of the proposed method is below that of the state-of-the-art methods. It would have been ideal if the authors had built their method on of the more recent high-quality nerf-based human avatar methods to achieve both overall high-quality and improvements in modeling of surface wrinkles.

---

> ### Author Response · Authors · 2023-11-21
> **Response to Reviewer McVe**
>
> We sincerely thank the reviewer for reading our paper and giving us thoughtful feedback. We carefully respond to each of the summarized concerns and questions. All these feedbacks will be incorporated into our revised manuscript.
>
> **Q1: The improvement in texture quality of the wrinkles are subtle.**
>
> We point out that it is more challenging to capture adaptive textures under different input poses when more frequency variations are presented in different video frames. There are two clips in the attached video for comparisons. Compared to the second clip where the highlighted pants region almost always presents wrinkles, the marked T-shirt region in the first clip shows wrinkles with much more variations and thus can better reveal our improvements in terms of wrinkle quality.
>
> We pick up two representative frames to highlight our wrinkle synthesis capability as it is difficult to directly evaluate the dynamic wrinkles. As exemplified in Fig. 1 and Fig. 10 of the submitted file, our model can faithfully enable pose-dependent details across different frames and pose contexts. Given that it is subjective to visually assess the produced wrinkles, we compute the matched histogram distances (**F-Dist**) between generated images and corresponding ground truths for quantitative comparisons. In Fig. 10-13, our proposed method can synthesize closer frequency distribution to the ground truth, exhibiting our conceptual advantages in producing adaptive textures, including wrinkles.
>
> **Q2: Marginal numerical improvement in the reported metrics.**
>
> In Tab. 2 of the main text, we outperform DANBO by ~0.4 dB in PSNR and 0.04 in SSIM with a comparable number of trainable parameters. According to the recent literature (e.g. Tab. 2 in Vid2Avatar and Tab. 3-4 in MonoHuman), this can have already been recognized as a significant improvement. When moving to more challenging scenarios of novel pose rendering, our empirical advantages become more remarkable: 0.41 (PSNR) and 0.05 (SSIM).
>
> Our effectiveness can be also demonstrated by the generality to the monocular video dataset. On the MonoPerfCap dataset, our method improves DANBO with about 0.45 dB in PSNR and 0.008 in SSIM for novel pose rendering.
>
> Lastly, the goal of our paper is to highlight the importance of pose modulated frequency modeling, which can also be demonstrated through the ablation studies. In Tab. 1 of the main text, our full model significantly outperforms the ablation baselines, clearly supporting the effectiveness of the pose modulated frequency learning.
>
> **Q3: Overall the results in the supplementary videos are quite blurry.**
>
> With the pose modulated frequency learning, our goal is to capture the desired textures with reduced artifacts for animated avatars. The sequences in the supplementary show that the proposed method can consistently outperform baselines with better preserved patterns in various scales. One possible explanation for the blurry issue is that the highlighted close-ups for comparisons have relatively low resolutions and the background is in black. Note that another visual artifact is the masks used to extract foregrounds introducing some jagged edges.
>
> **Q4: Geometry comparisons on modern baselines.**
>
> In the following table provides the geometry comparison results with the settings presented in the Sect. 4.2 of ARAH (Wang et al., 2022). Specifically, we also apply NeuS [1] to obtain the pseudo geometry ground truth and compute quantitative metrics with Chamfer Distance. Our method possesses better geometry reconstruction than the template-free methods including HumanNeRF and MonoHuman. The geometry scores are worse than Vid2Avatar which is a template based baseline and relies on SMPL parameters as a prior. The SDF-based surface rendering also helps Vid2Avatar achieve smoother shapes compared to our density-based learning.
>
> |      | Vid2Avatar | Vid2Avatar+mask | HumanNeRF | MonoHuman |  Ours  |
> |:----:|:------------:|:-------------:|:---------:|:---------:|:------:|
> | S377 |     0.1180    |     0.0336     |   0.0401  |   0.0897  | 0.0337 |
> | S387 |     0.4696    |     0.0283     |   0.0777  |   0.0801  | 0.0365 |
> | S393 |     0.3330    |     0.0404     |   0.0609  |   0.0836  | 0.0455 |
> | S394 |     0.2480    |     0.0410     |   0.0543  |   0.1006  | 0.0441 |
> |  Avg |     0.2921    |     0.0359     |   0.0582  |   0.0885  | 0.0400 |
>
> Besides the quantitative comparisons, we also report the visualizations in Fig. 14 of the updated paper. As a template-free method, HumanNeRF & MonoHuman introduce obvious floating artifacts in empty space while our method generates a much smoother body surface.
>
> Please note that our method focuses on how to capture high-frequency texture details for the view rendering. How to accurately reconstruct shapes is another exciting yet non-trivial research problem, and we leave it for future exploration.
>
> [1] NeuS: Learning Neural Implicit Surfaces by Volume Rendering for Multi-view Reconstruction, NeurIPS 2020.

---

### Author Response · Authors · 2023-11-21
**Common Response to All Reviewers**

We thank all reviewers for the detailed feedback and the time committed. The Reviewer recognizes the pose-dependent frequency modeling as an original new and conceptually sound idea (Re McVe). It also makes sense to introduce the details (Re SAAP) with a technically sound experimental methodology (Re McVe). The paper is well-written and structured (Re McVe, Re SAAP, Re PkgA) with superior performance (Re McVe, Re PkgA). The task to be solved is recognized to have high practical significance (Re SAAP). However, the limited evaluation on modern baselines is criticized.

We hope that the clarifications and additional results reported below, which demonstrate that the claims made in the paper hold over additional datasets and are temporally consistent, can alleviate the evaluation concern. To ensure reproducibility, we will make the codes of our method available upon acceptance.


**Q: Comparisons with modern baselines, like HumanNeRF, MonoHuman, NeuMan and Vid2Avatar.**


Thank you for the constructive comments. As suggested by Reviewer PkgA, we evaluate our method and three baselines (HumanNeRF, MonoHuman, and Vid2Avatar) over four full sequences (S377, S387, S393 and S394) of the ZJU-Mocap dataset to show our advantages with better generality to indoor monocular videos. Specifically, we follow the experimental settings of HumanNeRF and MonoHuman and achieve results through their public codes with default hyper-parameters. To perform a more reasonable evaluation, we apply SAM [1] to extract accurate foreground masks for training and inference.

We illustrate one comparison example in Figure 13 of the uploaded revision. As reported in the table below, our method achieves comparable or better quantitative results than all three baselines in terms of novel view synthesis which is suggested by Reviewer McVe. The results clearly demonstrate the effectiveness of the proposed pose-frequency modulations.

|      | NeuralBody | HumanNeRF | MonoHuman | Vid2Avatar | Vid2Avatar+mask |   Ours   |
|:----:|:----------:|:---------:|:---------:|:-------------:|:--------------:|:-----:|
| PSNR |    28.19   |   29.14   |   29.21   |     28.44     |      28.87     | **29.55** |
| SSIM |    0.958   |   0.965   |   0.966   |     0.938     |      0.967     | **0.968** |

Specifically, our method outperforms MonoHuman by ~ 0.35dB in PSNR and 0.003 in SSIM, but with much smaller network size: 3M (ours) vs 74M (MonoHuman). Similarly we achieve better results than HumanNeRF whose number of trainable parameters is ~60M. According to the recent literature (e.g. Tab. 2 in Vid2Avatar and Tab. 3-4 in MonoHuman), this can have already been recognized as a significant improvement. Please note that, **‘Vid2Avatar’** indicates the original Vid2Avatar implementation modeling both the human and the background in the scene jointly. In parallel, **‘Vid2Avatar+mask’** is the model only focusing on the foreground human representations, thus obtaining better metrics than the original version.

Due to the constraints of time and computation resources, we will incorporate the results of comparisons for novel pose rendering  into the revision. Additionally, note that, we outperform the follow-up works (e.g. Vid2Avatar and  MonoHuman) that in turn outperform NeuMan.

[1] Segment Anything, ICCV 2023.

---

> ### Comment · Reviewer_McVe · 2023-11-22
> **Thank you!**
>
> I have carefully read the authors' responses. I would like to thank the authors for diligently answering all my questions and for the additional experiments that they conducted. My doubts have been satisfied. I don't have any further questions. My outlooks towards this paper remains positive.

---

> ### Author Response · Authors · 2023-11-22
> **Thank you for your positive feedback!**
>
> Dear Reviewer McVe,
>
> We thank you very much for your positive and constructive feedbacks! We are especially glad that our qualitative and quantitative results as well as the improvements over baselines are acknowledged. We really appreciate your time and knowledge on guiding us to advance our paper from these key aspects.
>
> Best,
> The Authors

---

### Author Response · Authors · 2023-11-21
**Revision Uploaded**

We thank all of the reviewers for their time and effort in providing these helpful suggestions. Based on these reviews, we have raised our paper and uploaded the new version. The revised texts are highlighted in blue. For each reviewer, we provide a detailed, more specific response to their points, including our changes to the revised paper.

To summarize, we have made the following changes to our paper and supplementary material:

1. Adding one figure in Fig. 13 to visually compare with modern baselines on ZJU-Mocap examples.
2. Corresponding to [1], adding one table in Tab. 3 to quantitatively compare with modern baselines on ZJU-Mocap sequences.
3. Adding one figure in Fig. 14 to exemplify comparisons with modern baselines for geometry reconstruction.
4. Corresponding to [3], adding one table in Tab. 4 to quantitatively compare with modern baselines for geometry reconstruction.
5. In Section C of the appendix, adding a new paragraph block named “Modern baselines on ZJU-Mocap sequences.” to discuss the comparisons with state-of-the-art methods on the ZJU-Mocap dataset.
6. Updating the text of Sec. 4.5 to add discussions with the **noGNN** ablation baseline;
7. In Tab. 1, correcting the caption for (b) and adding one column for the **noGNN** ablation baseline;
8. In Fig. 7 (a), adding one sub-figure to visualize the result of the **noGNN** ablation baseline;
9. In Fig. 15, adding one figure to illustrate the detailed architecture of the **noGNN** ablation baseline;
10. In the attached video, inserting a new video section named **Temporal Smoothness Comparisons** to evaluate the temporal smoothness of our method.

---

> ### Author Response · Authors · 2023-11-22
> **Clarifications on “Vid2Avatar+mask”**
>
> We thank all reviewers for their input again. Besides our response submitted before, we add some clarifications on why we train **Vid2Avatar+mask** on ZJU-Mocap sequences.
>
> Joint learning of separating humans from arbitrary background and reconstructing detailed avatar surfaces is the key claim in the original Vid2Avatar paper. However, the background of ZJU-Mocap sequences is black and some parts of the foreground look dark, thus making it more challenging to accurately extract foreground and synthesize the texture details. Please see Fig. 13 and Fig. 14 of the uploaded revision as an example. After discussing with the authors of Vid2Avatar, we add the mask to impose the groundtruth supervision and explicitly extract the foreground as the **Vid2Avatar+mask** model. However, as shown in the comments in **Common Response to All Reviewers**, the pose-dependent frequency modulation successfully improves these strategies used in Vid2Avatar paper. Plus our model complexity and the results on geometry reconstruction, we have comprehensively revealed our advantages over the most recent baselines.
>
> Please let us know if you have further questions during the reviewer-author discussion period.

---

### Author Response · Authors · 2023-11-23
**Thanks for the reviews**

Dear reviewers,

As the discussion period is close to end, we thank you for your time and efforts on reviewing our paper with numerous constructive suggestions, from detailed expositions, technical motivations to comparisons with most recent methods. We really appreciate them! During the author-reviewer discussion period, the authors have tried to address reviewers’ every comment with great efforts. In Particular, we significantly consolidated our experiment part by adding comparisons with the baselines on ZJU-Mocap suggested by all reviewers. With the additional experiments, we can conclude with the effectiveness of the proposed pose-guided frequency modulation which is regarded to be interesting by Reviewer McVe and Reviewer PkgA till now. And the results are also acknowledged by reviewers. Finally, the authors promise to incorporate the ongoing experiments like novel pose rendering into the future revision.

---

### Meta-Review · Area_Chair_Ae5o · 2023-12-05

**Metareview:**

This paper proposes a method for template-free digital human reconstruction from videos. The key insight is to utilize frequency modulation to enhance detail learning (e.g., wrinkles in clothing). All reviewers recognize the paper as well-written, tackles a practical task and proposed an interesting idea. The primary concerns include a) subtle numerical improvement and b) missing comparisons to modern methods. During rebuttal, the authors have diligently provided more results to address these concerns.

**Justification For Why Not Higher Score:**

Though the idea is interesting and the results are solid, the contribution of this paper is limited. It largely builds on a prior work (i.e., DANBO, Su et al. 2022), and incorporate a frequency modulation module on top of it. Thus I am not recommending higher scores.

**Justification For Why Not Lower Score:**

The reviewers mainly raise two weaknesses: a) the numerical improvement is not significant compared to existing works and b) there is no comparison against more modern baselines. During the rebuttal, the authors have provided more qualitative results to demonstrate that the proposed method has the ability to reconstruct intricate details (e.g., wrinkles in clothing) and compared with baselines proposed by reviewers. Thus I would recognize the weaknesses have been adequately addressed and recommend the paper for acceptance.

---

### Decision · Program_Chairs · 2024-01-16

Accept (poster)